# Decreased maternal serum acetate and impaired fetal thymic and regulatory T cell development in preeclampsia

Mingjing Hu [1,2,24], David Eviston[2,24], Peter Hsu[3,4,24], Eliana Mariño [5], Ann Chidgey[6], Brigitte Santner-Nanan[1,2], Kahlia Wong[6], James L. Richards[5], Yu Anne Yap[5], Fiona Collier[7,8,9], Ann Quinton [2,10], Steven Joung[2,11], Michael Peek [2,12], Ron Benzie[11,13], Laurence Macia[14], David Wilson[15], Ann-Louise Ponsonby[9,16], Mimi L.K. Tang [9,17,18], Martin O'Hely[7,9], Norelle L. Daly[15], Charles R. Mackay [5], Jane E. Dahlstrom[19], The BIS Investigator Group[#], Peter Vuillermin[7,8,9,20,25] & Ralph Nanan[1,2,25]

Maternal immune dysregulation seems to affect fetal or postnatal immune development. Preeclampsia is a pregnancy-associated disorder with an immune basis and is linked to atopic disorders in offspring. Here we show reduction of fetal thymic size, altered thymic architecture and reduced fetal thymic regulatory T (Treg) cell output in preeclamptic pregnancies, which persists up to 4 years of age in human offspring. In germ-free mice, fetal thymic $CD4^+$ T cell and Treg cell development are compromised, but rescued by maternal supplementation with the intestinal bacterial metabolite short chain fatty acid (SCFA) acetate, which induces upregulation of the autoimmune regulator (AIRE), known to contribute to Treg cell generation. In our human cohorts, low maternal serum acetate is associated with subsequent preeclampsia, and correlates with serum acetate in the fetus. These findings suggest a potential role of acetate in the pathogenesis of preeclampsia and immune development in offspring.

[1] Charles Perkins Centre Nepean, The University of Sydney, Penrith 2750 NSW, Australia. [2] Sydney Medical School Nepean, The University of Sydney, Penrith 2750 NSW, Australia. [3] Discipline of Paediatrics and Child Health, Sydney Medical School, The University of Sydney, Sydney 2006 NSW, Australia. [4] Department of Allergy and Immunology, The Children's Hospital at Westmead, Sydney 2145 NSW, Australia. [5] Infection and Immunity Program, Biomedicine Discovery Institute, Monash University, Clayton 3800 VIC, Australia. [6] Department of Anatomy and Developmental Biology, Biomedicine Discovery Institute, Monash University, Clayton 3800 VIC, Australia. [7] Deakin University, Geelong 3220 VIC, Australia. [8] Barwon Health, Geelong 3220 VIC, Australia. [9] Murdoch Children's Research Institute, Parkville 3052 VIC, Australia. [10] School of Health, Medical and Applied Science, Central Queensland University, Sydney 2000 NSW, Australia. [11] Nepean Hospital, Penrith 2750 NSW, Australia. [12] ANU Medical School, College of Health and Medicine, The Australian National University, Canberra 0200 ACT, Australia. [13] Discipline of Obstetrics, Gynaecology and Neonatology, Sydney Medical School Nepean, The University of Sydney, Penrith 2750 NSW, Australia. [14] Department of Pathology, School of Medical Sciences, Charles Perkins Centre, The University of Sydney, Sydney 2006 NSW, Australia. [15] Centre for Molecular Therapeutics, AITHM, James Cook University, Cairns 4814 QLD, Australia. [16] National Centre for Epidemiology and Population Health, Research School of Population Health, College of Health and Medicine, The Australian National University, Canberra 0200 ACT, Australia. [17] The Royal Children's Hospital, Parkville, Melbourne 3052 VIC, Australia. [18] Department of Paediatrics, University of Melbourne, Melbourne 3010 VIC, Australia. [19] Anatomical Pathology, ACT Pathology, Canberra Hospital and ANU Medical School, College of Health and Medicine, The Australian National University, Canberra 0200 ACT, Australia. [20] Centre for Food and Allergy Research, Parkville 3052 VIC, Australia. [24]These authors contributed equally: Mingjing Hu, David Eviston, Peter Hsu. [25]These authors jointly supervised this work: Peter Vuillermin, Ralph Nanan. [#]A full list of consortium members appears at the end of the paper. Correspondence and requests for materials should be addressed to R.N. (email: ralph.nanan@sydney.edu.au)

The maternal and early in utero environment probably affects health and disease later in life. This is particularly true of non-communicable diseases such as allergy and autoimmunity[1]. Preeclampsia is a common pregnancy-associated disorder that is unique to humans. It is believed to develop due to a breakdown in maternal-fetal immune tolerance, as demonstrated by maternal immune alterations including reduced regulatory T (Treg) cells[2,3]. Maternal immune changes in preeclampsia are generally mirrored in the fetal immune system[4]. In turn, there is some evidence that preeclampsia is associated with higher rates of allergy[5,6] and cardiovascular disease[7–9] in offspring. This is suggestive of a common, environmental or physiological factor that influences both maternal and fetal immune systems.

Short chain fatty acids (SCFAs), mainly acetate, butyrate and propionate, are major metabolic products of the gut microbiota, mostly produced through bacterial fermentation of dietary fiber. SCFAs have potent anti-inflammatory effects, which probably relate to their role in the gut and systemic immune homeostasis[10,11]. Butyrate is mainly (70–90%) absorbed by the colonic epithelium, whereas propionate is mostly taken up by the liver[12], leaving acetate as the most prominent circulating SCFA. Serum acetate concentrations range from 70–170 μM[12–14], in contrast to propionate and butyrate with serum concentrations ranging from 1–13 μM[12]. Three recent murine studies also showed that dietary acetate and butyrate can protect against the development of food allergy, asthma, and autoimmune type 1 diabetes (T1D) through modulation of immune tolerance[15,16]. For example, acetate and butyrate from high-fiber diet increased Treg numbers through enhancing retinol dehydrogenase activity in CD103+ tolerogenic dendritic cells (DCs)[13] or directly via HDAC inhibition[14]. Maternal intake of dietary fiber/acetate during pregnancy protected offspring from allergic airways disease (AAD) correlated with markedly increased Treg function[15]. Young non-obese diabetic (NOD) mice, treated with a combination of modified starches that release microbial acetate and butyrate, were protected from developing autoimmune type 1 diabetes (T1D) in 90% of cases, resulting from the cessation of β-cell destruction and restored immune tolerance associated with expanded Treg cells, and reduced autoreactive CD4+ and CD8+ T cells[16]. In addition, dietary acetate has been found also to protect against cardiovascular disease and hypertension in mouse models[15,16].

Interestingly, there is some evidence suggesting maternal gut microbiota may influence the pathogenesis of preeclampsia, with probiotic use associated with a reduced risk of preeclampsia[17]. Additionally, a Mediterranean diet as well as a diet high in fiber, which promote SCFA production, are also associated with decreased preeclampsia[18,19]. Furthermore, in a murine model, supplementation of high fiber or direct supplementation of acetate significantly reduced both systolic and diastolic blood pressures, a finding highly relevant to preeclampsia with hypertension as a defining clinical feature[20].

Maternal microbial ecology is known to influence fetal immune development[21]. In mice, several studies have shown that maternal microbial composition is important for the protection against the development of allergic disease in the offspring[22,23] and can influence the infant immune response[24]. In humans, various cross-sectional and cohort studies have confirmed these findings and shown that maternal microbial exposure (for example in the farm environment) is associated with changes in offspring immune profiles and protection against atopic disorders[25,26]. Exactly how this protection is conferred is unknown; however, a role for Treg cells[27] and Toll ligand receptors (TLR), which can be activated by bacterial products[27], has been inferred. Therefore, an alluring concept is that levels of production of SCFAs by maternal gut microbiota may influence both maternal and fetal immune homeostasis during pregnancy.

In this study, we present evidence of an association between maternal gut microbial metabolites and fetal immune development in preeclampsia. Specifically, we demonstrate in mice a link between acetate and fetal thymic development and output, with concordant associations in human cohorts between decreased serum acetate and subsequent preeclampsia, and between preeclampsia and decreased thymic size and output in the offspring. Maternal acetate supplementation had a significant impact on the fetal immune system in a germ-free mouse model, with offspring demonstrating increased CD4+ T-cell production, improved Foxp3 expression and rescue of specialized thymic epithelial cells expressing the AIRE gene, essential for self-tolerance induction and Treg cell generation early in life[28].

## Results

**Fetal thymic development is compromised in preeclamptic pregnancies**. We have shown previously that reduced fetal thymic size is evident in preeclamptic pregnancies by examining fetal ultrasound images in retrospectively assembled case–control series[29]. Here, we performed prospective studies in two separate cohorts. First, we compared fetal thymus volume and diameter measurements between 50 preeclamptic and 50 non-preeclamptic pregnant women (*Nepean cohort 1*, patient characteristics are shown in Supplementary Table 1a). Then in a second cohort, we recruited 887 pregnant women between 17- and 22-week gestation, of whom 24 developed preeclampsia later in their pregnancies (*Nepean cohort 2*, patient characteristics are shown in Supplementary Table 2).

Maternal and fetal characteristics in Nepean cohort 1 were similar between the groups, except preeclampsia was associated with increased rate of nulliparity, higher maternal body mass index (BMI; measured at the first antenatal visit), a greater average estimated fetal weight and earlier gestation at delivery (Supplementary Table 1a). Antenatal ultrasound scans showed that, compared to normal pregnancy, preeclampsia was associated with a 38% decrease in fetal thymic volume (1.6 versus 2.6 mL, $p < 0.001$ (unpaired *t*-test), Supplementary Table 1b) and 12.1% decrease in fetal thymic diameter (2.9 versus 3.3 cm, $p = 0.01$ (unpaired *t*-test), Supplementary Table 1b). Evidence of difference persisted following adjustment for estimated fetal weight percentile, maternal BMI, and smoking status ($p = 0.04$ and $p = 0.009$ for mean fetal thymus volume and diameter (univariate logistic regression), respectively). The differences in thymus volumes between preeclampsia and healthy controls were small at early gestation but increased linearly over gestation (Fig. 1b). By term, the average thymus volume in fetuses born to preeclampsia women was approximately half of that in fetuses born to normal pregnancies. For each 1 mL increase in volume, the mean thymus diameter (MTD) increased on average by 1.03 cm (95% CI: 0.79–1.28, $p < 0.001$ (univariate logistic regression), R2 = 41%).

In Nepean cohort 2, each pregnancy had fetal thymic measurements performed at mid-gestation between 18 and 22 weeks, long before the onset of clinical preeclampsia. Once again, preeclampsia was associated with smaller fetal thymus diameter (Fig. 1d); adjusted means were 16.5 versus 18.3 mm, respectively ($p < 0.001$ (unpaired *t*-test), Supplementary Table 2). The odds of preeclampsia increased by 1.41 (1.17, 1.69) for each 1 mm decrease in fetal thymus diameter ($p < 0.001$, univariate logistic regression). Adjusting for gestational age and maternal BMI, multivariate logistic regression (Model 1) found the odds of preeclampsia increased by 1.47 (1.22, 1.78) for each 1 mm decrease in fetal thymus diameter ($p < 0.001$, univariate logistic regression). The magnitude and evidence of association also

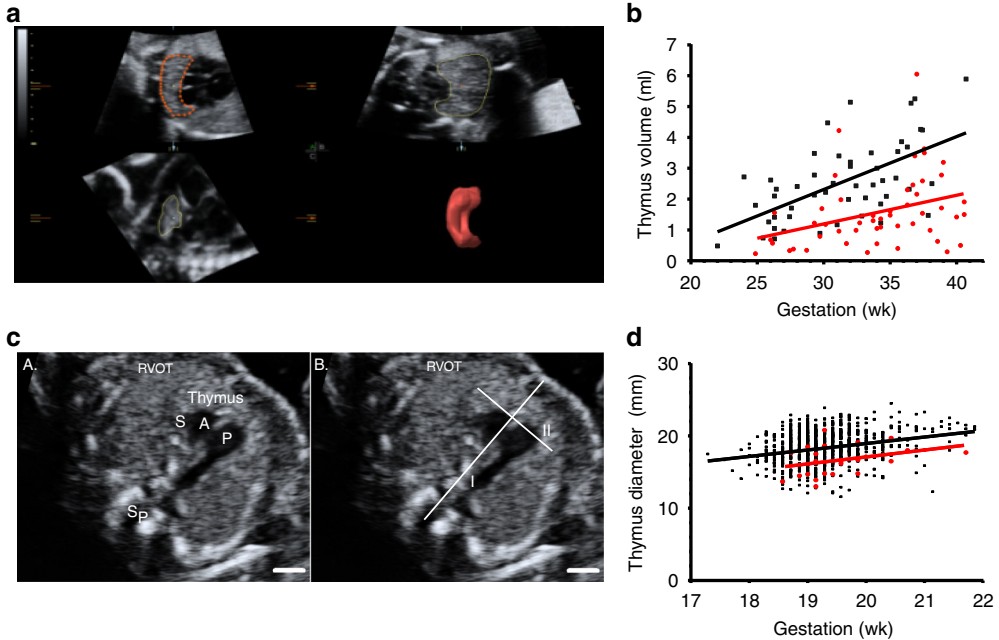

**Fig. 1** Impaired fetal thymic development in preeclampsia. **a** *Nepean cohort 1*: Virtual organ computer-aided analysis calculation of the fetal thymus volume at 26 weeks of gestation, showing three orthogonal planes (left top: transverse, right top: coronal, left down: sagittal) and reconstructed thymus volume (right down). **b** Dot plot showing fetal thymus volumes in different gestational weeks in both non-preeclamptic ($n = 50$, black dots) and preeclamptic ($n = 50$, red dots) groups. Mean ± SD: 2.6 ± 1.3 mL and 1.6 ± 1.2 mL, respectively ($p < 0.001$, unpaired $t$-test). **c** *Nepean cohort 2*: Fetal thymus diameter measurement: an axial view of the fetal thymus was obtained, within a standard image of the right ventricular outflow tract (RVOT). Then a line was drawn connecting the fetal spine and sternum (I). Fetal thymus diameter was measured as its greatest diameter (II), perpendicular to line I. Sp = spine, S = superior vena cava, A = aorta, P = pulmonary artery. Scale bar = 5 mm. **d** Dot plot showing fetal thymus diameter in different gestational weeks in both non-preeclamptic ($n = 863$, black dots) and preeclamptic ($n = 24$, red dots) groups, adjusted means were 18.3 mm and 16.5 mm, respectively ($p < 0.001$, unpaired $t$-test)

persisted following adjustment for fetal head circumference and maternal BMI (fetal thymus diameter odds ratio = 1.51 (1.23, 1.85, $p < 0.001$, univariate logistic regression).

Receiver operating characteristic (ROC) curves were generated to evaluate fetal thymus diameter at mid-gestation as a predictive tool for preeclampsia. Fetal thymus diameter at mid-gestation alone was found to have an area under the curve (AUC) of 0.71. Together with gestational age and maternal BMI, AUC increased to 0.78. Since our group had previously found that increased fetal head growth was also associated with preeclampsia after adjusting cofounders[30], we then included the fetal head circumference in the model. When fetal thymus diameter at mid-gestation, fetal head circumference at mid-gestation, gestational age, and maternal BMI were included, we could reach an AUC of 0.81 (Supplementary Fig. 1).

**Fetal thymic Treg cell output is reduced in preeclampsia.** We showed previously that cord blood and maternal blood Treg cell percentages were highly correlated[4], suggesting a common mechanism for Treg control in both mother and fetus. Here, we extended this finding and showed a correlation between maternal and fetal Treg cell proportions in both healthy pregnancy and preeclampsia. Patient characteristics (*Nepean cohort 3*) are shown in Supplementary Table 3. Peripheral maternal Treg correlated with cord blood Treg, but to a lesser extent in preeclampsia compared to non-preeclamptic pregnancies and overall Treg frequency was lower in preeclamptic dyads compared to non-preeclamptics (Y intercept at $X = 0$: 2.68 ± 0.56 versus 2.51 ± 0.92, $p = 0.019$, unpaired $t$-test) (Fig. 2a).

Maternal blood Tregs (the percentage of Foxp3+ cells within CD4+ cells) were lower in preeclampsia ($p = 0.004$, unpaired $t$-test) and similarly, there were some evidence of cord blood Treg

cell frequencies being lower in offspring of preeclamptic versus non-preeclamptic pregnancies ($p = 0.09$, unpaired $t$-test) (Fig. 2b). A subgroup of *Nepean cohort 3* that included 21 non-preeclamptic and 19 preeclamptic samples was selected based on access to stored samples, and we performed further phenotypic Treg cell analysis. We found that frequencies of cord blood CD4+Foxp3+Helios+ thymic-derived natural Tregs (nTreg) were also significantly lower in preeclampsia ($p = 0.035$, unpaired $t$-test), especially in those who did not receive steroid treatment ($p = 0.01$, one-way ANOVA with Dunnett's multiple comparisons test) (Fig. 2d). Altogether these findings show that offspring of preeclamptic mothers have reduced thymic volume and diameter with an associated deficit in the output of thymic Treg.

**Reduced number of naive Treg cells persist into early childhood.** To further examine whether the cord blood changes in preeclampsia were persistent, we examined the CD4+CD45RA+Foxp3+ thymic-derived naive Treg cells at birth, 6 and 12 months and 4 years of age from the BIS cohort, an Australian birth cohort study ($n = 1064$ mothers/1074 infants[31], patient characteristics are shown in Supplementary Table 4). There was some evidence that the proportion of naive Treg cells in cord blood was lower in infants born to mothers with preeclampsia ($p = 0.057$, linear regression) (Fig. 3b), and this association was stronger following adjustment for exposure to mode of delivery and gender ($p = 0.025$, linear regression), factors that have been shown to influence the proportion of cord blood naive Treg cells[32]. Longitudinal analysis revealed a persistent deficit in naive Treg proportion in offspring of preeclamptic mothers over the first 4 years of postnatal life ($p = 0.004$, generalized estimating equation) (Fig. 3b).

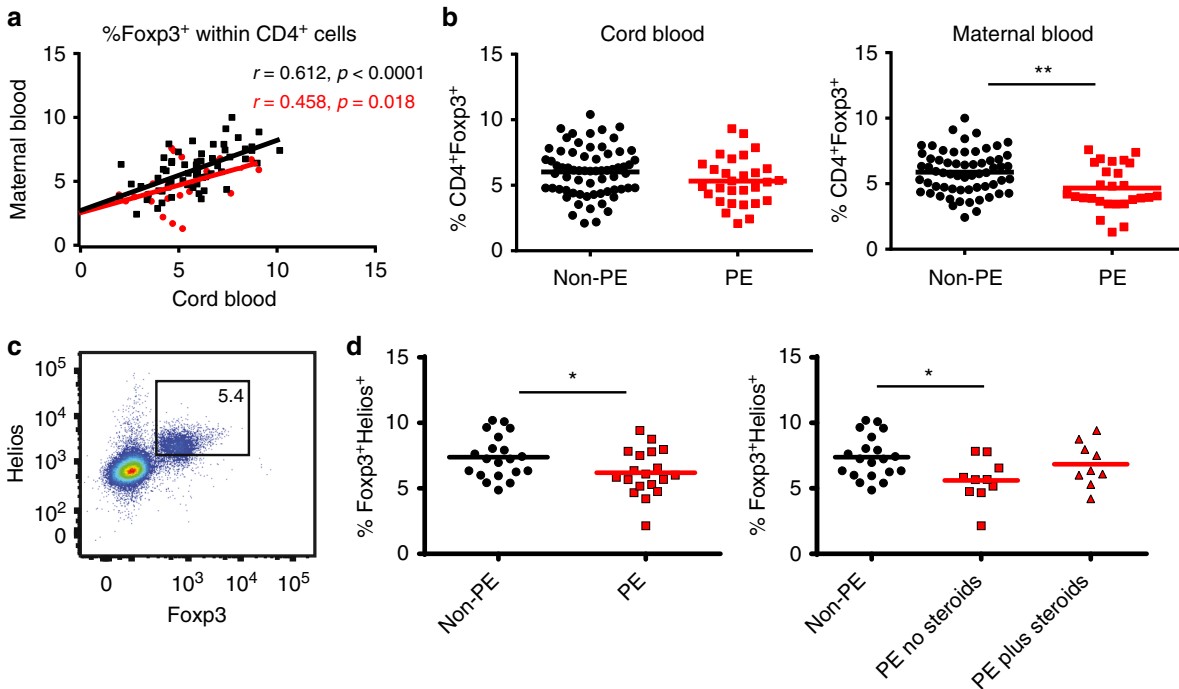

**Fig. 2** Reduced number of fetal Treg cells and fetal Treg thymic output in preeclampsia. *Nepean cohort 3*: **a** Scatter dot plot showing Treg cell frequencies in term maternal-fetal dyads in both non-preeclamptic (non-PE) ($n = 62$, black dots) and preeclamptic (PE) pregnancies ($n = 26$, red dots). Pearson's correlation was used. Coefficient $r$ and $p$-values as indicated. **b** Scattered dot plots comparing the %Foxp3+ (Mean) within CD4+ cells in cord blood ($n = 66$ for non-PE, $n = 29$ for PE group) and maternal blood ($n = 62$ for non-PE group, $n = 27$ for PE group) (unpaired $t$-test). Gating strategy for Foxp3+ Tregs from PBMC is shown in Supplementary Fig. 2. **c** Representative dot plot showing cord blood Foxp3+Helios+ cells gated on CD4+ cells. **d** Scattered dot plots showing the percentage of Foxp3+Helios+ cells (Mean) in cord blood. Left panel comparing %Foxp3+Helios+ cells in non-PE ($n = 20$) and PE ($n = 19$) groups (unpaired $t$-test). Right panel dividing the PE group into two subgroups based on steroids treatment: $n = 10$ for no Steroids and $n = 9$ for plus Steroids (one-way ANOVA with Dunnett's multiple comparisons test). *$p < 0.05$, **$p < 0.01$

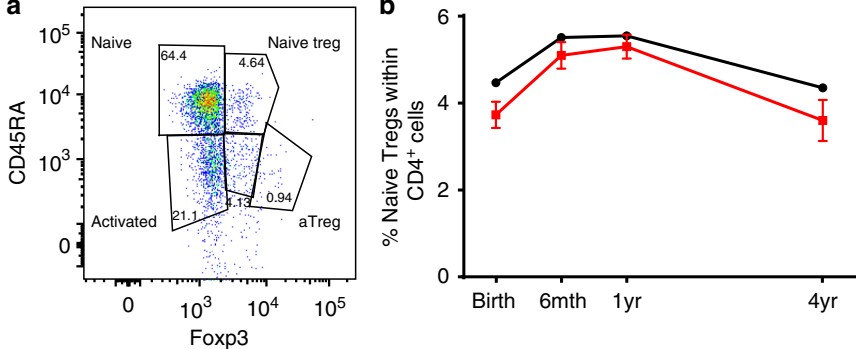

**Fig. 3** Naive Treg cell changes in infants exposed to preeclampsia. *BIS study*: **a** Representative dot plot showing the gating strategy for naive Tregs gated on CD4+ cells. **b** The line graphs showing the longitudinal naive Treg cell proportions (Mean ± SD, as a percentage of the CD4+ population) in children from preeclamptic mothers (Red: at birth $n = 11$; 6-month $n = 19$; 1-year-old $n = 22$; 4-year-old $n = 10$) and children from non-hypertensive mothers (Black: at birth $n = 434$; 6-month $n = 558$; 1-year-old $n = 629$; 4-year-old $n = 377$). The overall difference is –0.49% (95% CI –0.83, –0.16) ($p = 0.004$, generalized estimating equation)

**Reduced thymic cortex and T-cell numbers on autopsy in preeclampsia.** Thymuses of preeclamptic cases from the *Fetal autopsy* cohort (patient characteristics are shown in Supplementary Table 5a), showed features of grade 2–3 thymic involution with loss of the normal distinction between cortex and medulla observed at low magnification owing to advanced lymphophagocytosis and resulting in an irregular narrowing of the cortex, foci of lymphodepletion and increased separation of thymic lobules (Fig. 4). These are features commonly seen in association with chronic intrauterine hypoxia that is seen in preeclampsia[33]. Moreover, for the preeclampsia group, no

staining of Foxp3 was observed in these tissues. Conversely, in both thymus and spleen, 4 out of 6 non-preeclamptic group fetuses demonstrated Foxp3+ staining in <15% of cells. In the thymuses of fetuses born to non-preeclamptic mothers, Foxp3+ expression was mainly limited to the medulla.

In all fetuses of non-preeclamptic cases, thymic CD4+ T cells were predominant in the cortex, versus the medulla. However, in 4 out of 5 fetuses of preeclamptic cases, CD4+ T cells were equally prevalent in thymic cortex and medulla. Within the spleen, CD4+ T-cell expression was reduced in fetuses of preeclamptic versus non-preeclamptic mothers (Supplementary Table 5b). Taken

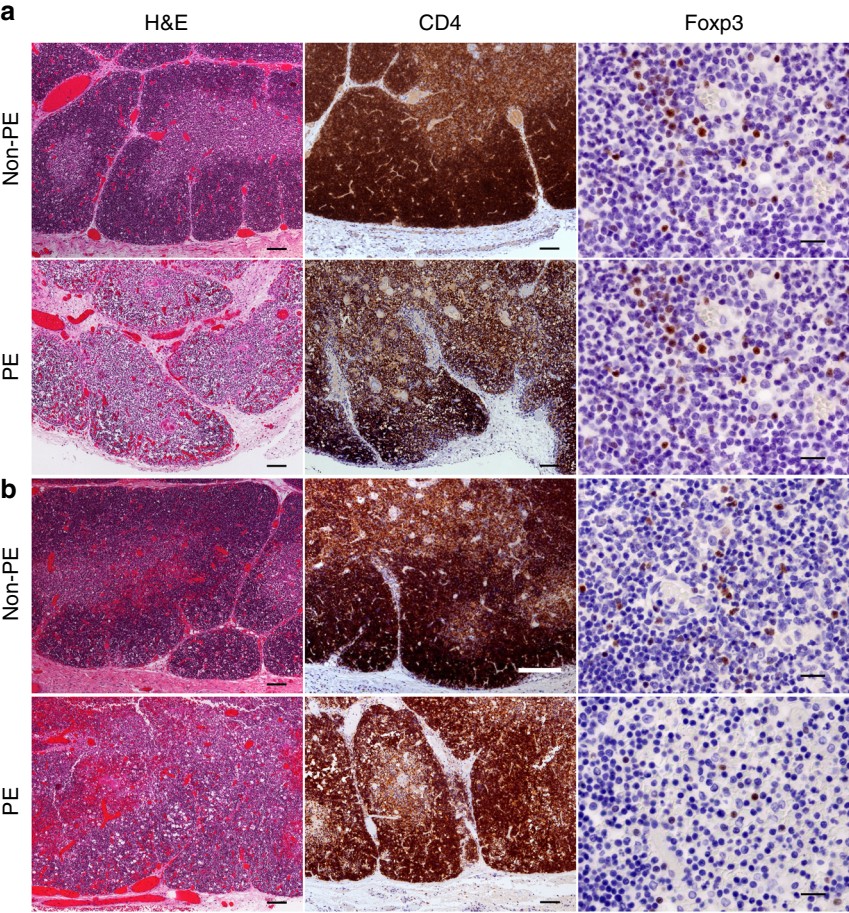

**Fig. 4** Fetal thymus histopathology. Photomicrographs of fetal thymuses stained to compare CD4+ and Foxp3+ T-cell populations between the control group and those mothers with preeclampsia at the time of fetal demise. **a** Fetal thymuses of Control (upper panel) and preeclamptic mothers (lower panel), at 24 weeks gestation. **b** A 33-week-old control fetus (upper panel), versus a 37-week-old fetus deceased to a mother who had preeclampsia (lower panel). Scale bars for H&E stains and CD4 stains = 100 μm. Scale bars for Foxp3 stains = 20 μm

together the above findings consistently demonstrate that preeclampsia is associated with thymic involution and reduced Treg expansion in offspring.

**Serum acetate in healthy pregnancy and preeclampsia.** Given the evidence of the association between gut microbiota and preeclampsia[17–19], we then examined the metabolites of gut microbiota—SCFAs—in maternal serum at 28 weeks gestation and its relationship with subsequent preeclampsia in the BIS cohort. Consistent with previous findings, serum acetate levels were at least tenfold higher compared to butyrate and propionate (Fig. 5a). On weighted logarithmic regression analysis, there was also evidence of associations between lower serum SCFAs and subsequent preeclampsia (Fig. 5b–d). The relative risk of developing preeclampsia was decreased per 30% increase in the serum acetate (RR 0.77, 95% CI 0.61–0.96, $p = 0.021$, weighted logarithmic regression). This evidence persisted following adjustment for maternal age ($p = 0.031$, weighted logarithmic regression). In a smaller subset in which pre-pregnancy BMI had been recorded (24 with preeclampsia, 253 without), the evidence of an association between serum acetate and preeclampsia was weak ($p = 0.178$, logarithmic regression) but minimally changed by adjustment for pre-pregnancy BMI ($p = 0.143$, logarithmic regression). Pre-pregnancy BMI was associated with preeclampsia ($p = 0.003$, logarithmic regression) but there was no evidence that this association was mediated by serum acetate (proportion of total effect mediated −3%, 95% CI −55 to 20%). The evidence of a univariate

association between serum acetate and preeclampsia was also attenuated by the exclusion of one older mother with low serum acetate and postnatal preeclampsia (RR 0.46, 95% CI 0.20–1.06, $p = 0.067$, logarithmic regression). In summary, an inverse association between lower serum acetate and subsequent preeclampsia was observed in the overall cohort with and without adjustment for maternal age but in a smaller subset, where pre-pregnancy BMI values were available this association did not reach significance. There was no evidence that maternal serum acetate measured at 28 weeks gestation, was positively associated with cord blood naive Treg. In the *Nepean cohort 4* of paired maternal and fetal serum (patient characteristics are shown in Supplementary Table 6), collected no longer than 4 h apart, we did find a correlation ($r = 0.367$, $p = 0.012$, Pearson's correlation) between cord blood and maternal blood acetate levels (Fig. 5e). This suggests that maternal acetate may cross the placenta and therefore influence fetal acetate levels. This is supported by our previous work, which showed that in mice acetate does indeed cross the placenta and affect concentrations in the developing fetus[14].

**Maternal acetate rescues fetal thymus in germ-free mice.** Mounting evidence suggests that SCFAs and other bacterial metabolites influence T cell/Treg cell development[14,34–37]. Therefore, we investigated whether maternal bacterial load could affect fetal thymic development by comparing thymus weight of pups (21-days-old) born to germ-free (GF) mice with those born to specific pathogen free (SPF) mice. Thymic weight (Fig. 6a) and

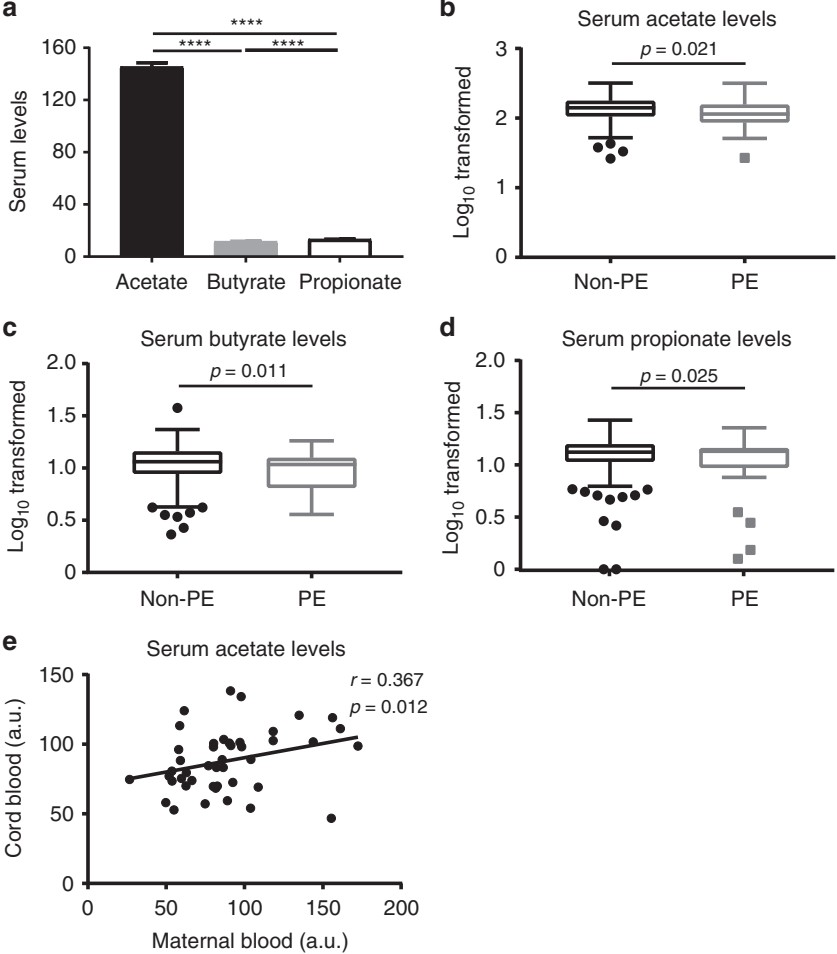

**Fig. 5** Serum acetate in healthy pregnancy and preeclampsia. *BIS study:* **a** Bar graph showing maternal serum levels (Mean ± SEM, µM) of acetate, butyrate, and propionate at 28 weeks of pregnancy ($n = 324$) (RM one-way ANOVA with Tukey's multiple comparisons test, with the Greenhouse-Geisser correction). Box and whisker (Tukey) plots showing evidence of associations between lower (**b**) acetate, (**c**) butyrate, and (**d**) propionate levels (µM, Log base 10 transformed) from women with subsequent preeclampsia (PE, $n = 31$) and women without preeclampsia (non-PE, $n = 293$). *p*-values calculated using weighted logarithmic regression. *Nepean cohort 4* (**e**): Scattered dot plot showing the relative serum acetate levels in paired term maternal peripheral blood and cord blood ($n = 46$) (samples were collected at full term delivery). The acetate levels were based on peak heights from the NMR spectra normalized to the total intensity. a.u. = arbitrary units (values are divided by 1000). Pearson $r = 0.367$, $p = 0.012$; Spearman rho = 0.416, $p = 0.004$

thymocyte cellularity (Fig. 6b) in GF mice was approximately half that of SPF mice, which is similar to our findings in preeclampsia, which we found to be associated with low acetate levels. Since our human cohort showed that serum acetate levels were tenfold higher compared to butyrate and propionate, we then supplemented the GF pregnant mice with acetate in the drinking water and analyzed pups. Interestingly, maternal acetate supplementation rescued both thymic weight and thymocyte cellularity in pups to levels comparable to SPF mice (Fig. 6a, b). This suggests that maternal gut microbiota-derived acetate contributes to fetal thymic development. Thymic progenitors are bone marrow-derived and in addition to the lower thymic cellularity, GF mice also had significantly reduced bone marrow cellularity (Fig. 6c); in particular, lymphoid-primed multipotent progenitors (CD62L$^+$ LMPP; Fig. 6d), which supply the thymus with progenitor cells for T-cell development. Maternal acetate supplementation improved both bone marrow cellularity and CD62L$^+$ LMPP cell proportions in pups (Fig. 6c, d), in parallel with thymic cellularity normalization.

**Maternal acetate potentiates fetal T-cell development.** We then compared the development of T-cell subsets in GF and SPF

fetuses. Offspring of GF mice had decreased total CD4$^+$ T-cell numbers (Fig. 6e) and fewer CD4$^+$ Foxp3$^+$ Tregs (Fig. 6f). In addition, reduced abundance of Foxp3 protein per cell, measured by mean fluorescence intensity (MFI), was apparent in GF mice (Fig. 6g). Interestingly, maternal acetate supplementation restored the total number of CD4$^+$ T cells in pups to that of SPF mice. While Treg numbers did not increase, acetate supplementation rescued Foxp3 MFI in Treg cells (Fig. 6g), with implications in Treg function. Altogether these results suggest that bacteria-derived acetate is important for fetal CD4$^+$ T-cell and Treg cell development and dietary acetate supplementation can correct defects resulting from an absent microbiota.

**Maternal acetate recovers offspring thymic AIRE expression.** AIRE contributes to Treg cell generation early in life[28] and Foxp3 levels are important for Treg function[38], therefore we investigated for alterations in AIRE expression in thymic epithelial cells. AIRE is mostly expressed in thymic epithelial cells[39]. Our gating strategy for flow cytometry was based on the expression of europaeus agglutinin 1 (UEA1) lectin, which discriminates cortical thymic epithelial cells (cTEC; UEA1$^-$) from medullary epithelial cells (mTEC; UEA1$^+$) while both subsets express major

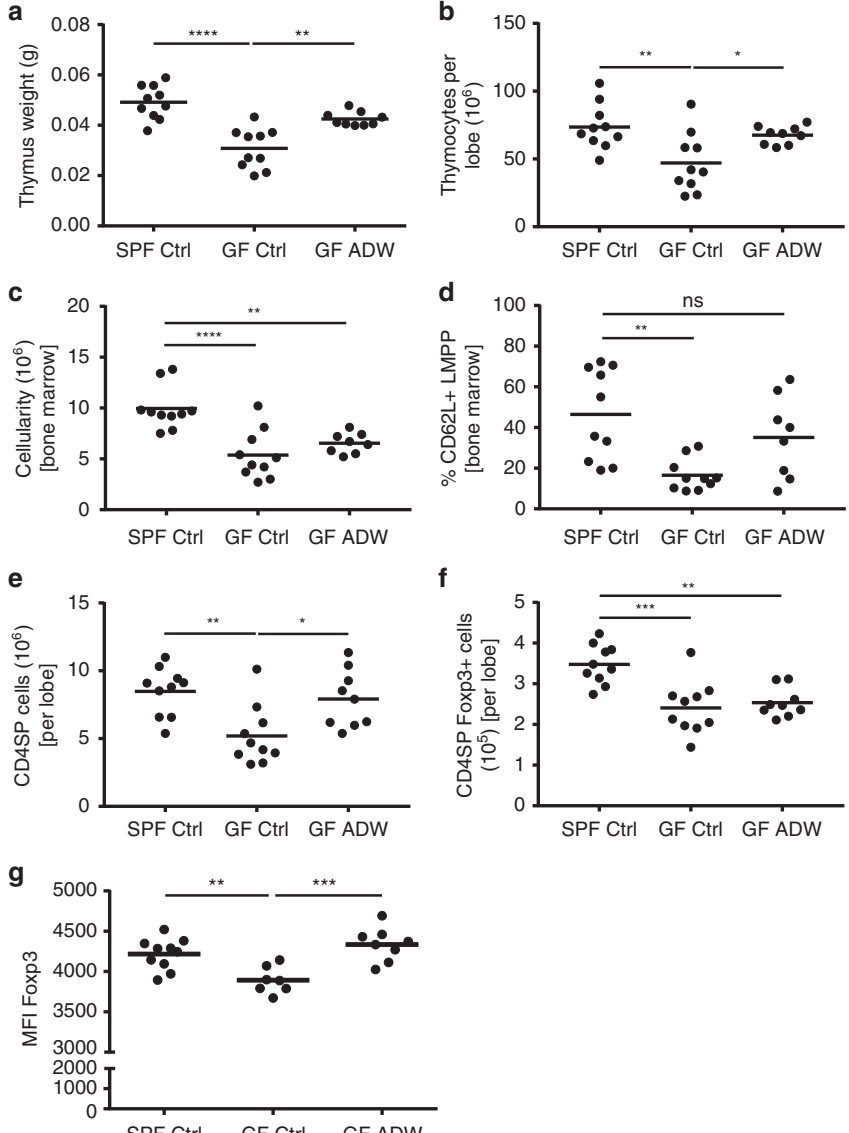

**Fig. 6** Reduced Foxp3 MFI in thymic Treg cells from GF pups recovered with acetate supplementation. Three-week-old C57BL/6 pups from control SPF and GF mothers and GF mothers treated with acetate in drinking water (ADW). Cumulative data showing phenotypic analysis by flow cytometry in the thymus from pooled female and male pups ($n = 7$–10/group). **a** Thymus weight. **b** Thymocyte cellularity per thymic lobe. **c** Bone marrow cellularity. **d** The frequency of bone marrow CD62L+ lymphoid-primed multipotent progenitor (LMPP) cells. **e** The total number of CD4SP cells. **f** The total number of CD4SP Foxp3+ cells. **g** Foxp3 protein expression on a per cell basis in thymic CD4SP Foxp3+ cells. Gating strategies for LMPP cells, CD4SP cells and Foxp3+ cells are shown in Supplementary Figs. 3 and 4. The data are shown as mean fluorescence intensity (MFI). *$p < 0.05$, **$p < 0.01$, ***$p < 0.001$, ****$p < 0.0001$. All data were analyzed by one-way ANOVA with Bonferroni's multiple comparisons test. Data represent mean; each symbol represents an individual mouse. Data are representative of at least two independent experiments

histocompatibility complex II (MHC II) (Fig. 7a and Supplementary Fig. 5a). The majority of AIRE+ cells are found in the keratin 14+ medulla region (Supplementary Fig. 5b). However, we also found a minority of AIRE+ cells with a cortical phenotype (β5t+, keratin 14−) situated at the cortico-medullary junction (white arrows, Supplementary Fig. 5b). We found a significant reduction in the number and proportion of AIRE expressing cells in the cTEC subset in GF mice while AIRE-positive cells among the mTEC were unchanged (Fig. 7d, e), suggesting a differential role for gut bacterial metabolites on AIRE expression in these subsets. The proportion of AIRE expressing cTEC was normalized in the offspring of mice supplemented with acetate during pregnancy (Fig. 7b, c). Therefore, the impact of acetate on AIRE+ cTEC population might contribute to the effect of acetate on Foxp3 expression in Tregs.

## Discussion

These studies provide a plausible link between the maternal intestinal bacterial metabolite acetate, thymus, and Treg cell development in the context of preeclampsia, which is a pregnancy disorder associated with important adverse outcomes for both the mother and their offspring. We definitively demonstrated an association between preeclampsia and fetal thymic hypoplasia. Fetal thymus diameter at mid-gestation might have clinical utility as a predictive tool for preeclampsia. This is illustrated by ROC curve analysis, which returned an AUC of 0.81 for a model consisting of fetal thymus diameter at mid-gestation, fetal head circumference at mid-gestation, gestational age, and maternal BMI. Such an early, non-invasive predictor of preeclampsia could have significant clinical value; however, the utility of this index would require further evaluation.

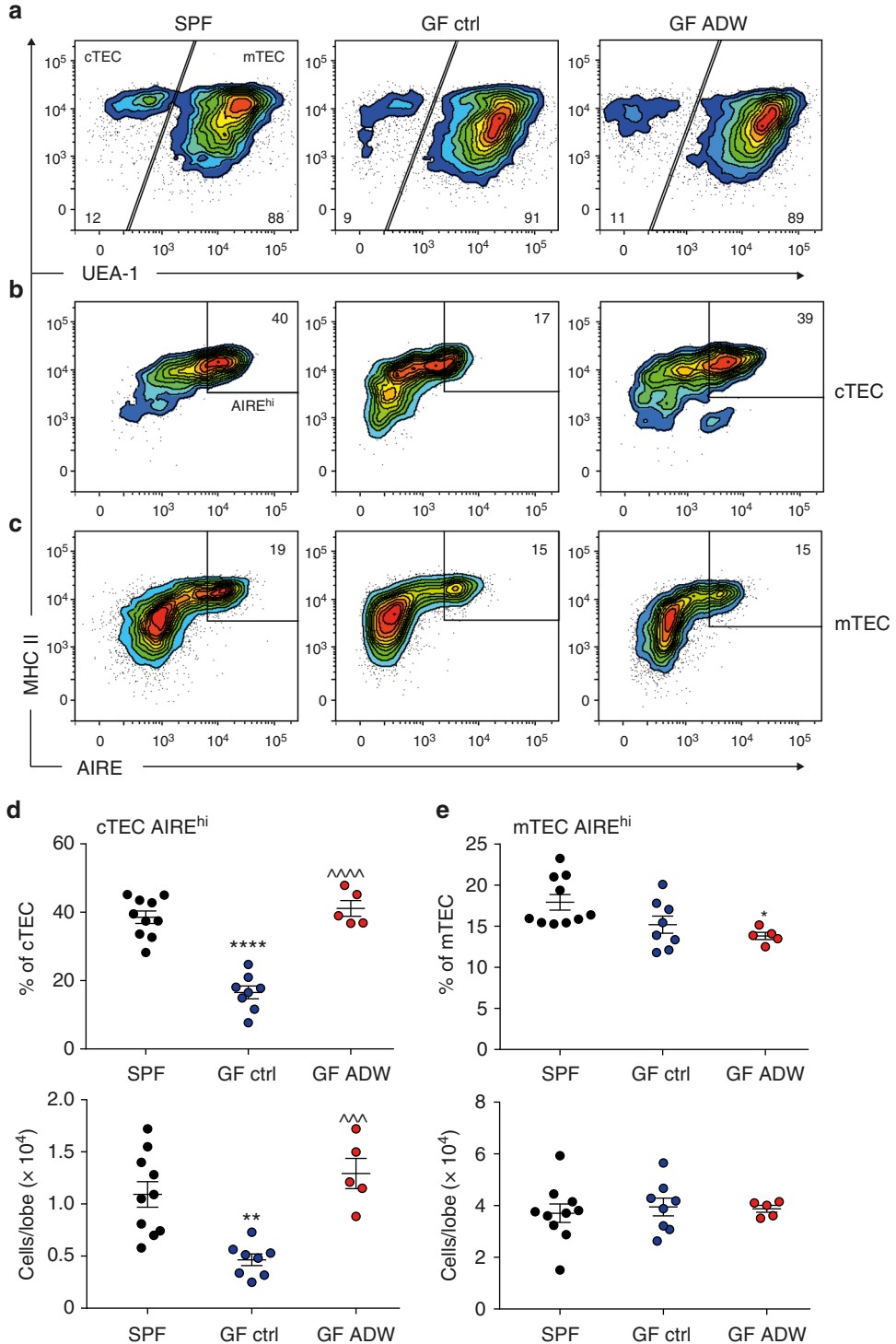

**Fig. 7** Reduced thymic AIRE expression in GF pups recovered with acetate supplementation. Three-week-old C57BL/6 pups from control SPF and GF mothers and GF mothers treated with acetate in drinking water (ADW). **a–c** Representative flow cytometry analysis demonstrating AIRE expression in thymic epithelial cell subsets. **d, e** Flow cytometry and numerical analysis of AIRE expression in cTEC and mTEC subsets from pooled female and male pups ($n = 5–10$/group). Asterisk denotes significance to SPF, $*p < 0.05$, $**p < 0.01$, $****p < 0.0001$; ^denotes significance to GF control, $\hat{}\hat{}\hat{}p < 0.001$, $\hat{}\hat{}\hat{}\hat{}p < 0.0001$. All data were analyzed by one-way ANOVA with Bonferroni's multiple comparisons test. Data represent mean; each symbol represents an individual mouse. Data are representative of at least two independent experiments

Furthermore, we showed the impact of impaired fetal thymic development with reduced Treg cell output in the offspring of preeclamptic mothers, which was evident at birth and persisted into early childhood. We then showed a reduction in Treg cell frequency in preeclamptic mother-infant dyads compared to

healthy pregnancies, consistent with a deficit in an immune regulatory factor that is able to cross the placenta. Further examination of cord blood Treg cells revealed that the deficiency mainly lies in the resting and thymus-derived Treg cells, implicating reduced thymic Treg cell output, which is also consistent

with absent Foxp3+ cells in thymic sections of deceased fetuses of preeclamptic mothers. We found no evidence that preeclamptic pregnancies were associated with changes in percentages of fetal thymic T cells (CD3 T cells), or conventional CD4/CD8 T-cell subsets (Supplementary Fig. 6). We have no data on the absolute (total) numbers of thymic Treg cells or CD4/CD8 T-cell subsets. To calculate total numbers of these subsets would have required the availability of full blood counts, which were not systematically collected for any of the cohorts. However, there is compelling data suggesting that changes in percentages of Treg cells within CD4+ cells play a central role in many immune-mediated pathologies[40–43]. In line with this, our own group has shown reduced percentages in maternal Tregs to be a feature of altered immunity in preeclampsia[4,44,45]. Hence, differences in percentages of offspring Treg cells described in this study between the preeclampsia and non-preeclampsia groups seems highly relevant. Importantly, we showed that fetal thymic changes precede clinical preeclampsia suggesting that these deficits in fetal immune development may be a component of the primary pathology driving preeclampsia, rather than a consequence of physiological stress associated with the clinically manifest disease.

Many factors may affect fetal immune development. Previously we showed that IL-10 is likely to be involved in the fetal-maternal alignment of Treg cells[4]. However, while IL-10 levels are reduced in preeclampsia[46], there is little to support its role in fetal thymic development. By contrast, mounting evidence implicates the role of microbial production of SCFAs in shaping both the innate and adaptive immune system[47]. Interestingly, changes in the oral and placental microbiota have been observed in preeclampsia[48,49] and a high-fiber diet appears to be protective against the development of preeclampsia[19], consistent with a role for diet-microbiota interplay in the prevention of preeclampsia. Moreover, the fact that acetate also has a direct effect on systemic blood pressure[20], implies beyond its immune modulating effects, a special role of this SCFA in preeclampsia with hypertension as its defining clinical feature[20].

In view of this, we sought to test whether maternal gut microbiota metabolites might influence fetal thymic immune development. In a mouse model, we found a key role for maternal microbiota on lymphoid organs and immune cell development. We show that the maternal acetate can affect the development of both thymus and bone marrow in the offspring as well as CD4+ T-cell development and the expression of AIRE and Foxp3, key components for Treg function. The absence of a maternal gut microbiota led to the offspring with reduced BM-derived LMPP precursors that may underpin the substantial reduction in thymic size and thymic output. This could be rescued by supplementing maternal drinking water with acetate, which is the most abundant SCFA in human serum[12,50,51]. Decreased Treg cell Foxp3 levels were also found coincident with a reduction in AIRE expression in specialized thymic epithelial cells in the absence of maternal gut microbiota. Both were rescued with maternal acetate supplementation. Treg cell generation is influenced by levels of AIRE expression in medullary thymic epithelial cells[52] and their functional status is reflected by the levels of Foxp3[38]. While the majority of AIRE expression is found deep within the thymic medulla, we identified a small proportion of AIRE expressing mature cortical thymic epithelial cells located at the corticomedullary junction that was influenced by maternal gut microbiota. Our findings that a reduction in this specialized AIRE+ epithelial cell population is associated with the levels of Foxp3 expression in Treg cells, suggests a role in Treg functionality. While AIRE expressing medullary epithelial cells express specialized tissue-restricted self-antigens, these cortical epithelial cells may present abundant, ubiquitous self-antigens early in Treg development[53]. Thus, recovery of AIRE expression in cortical

epithelial cells and increased Foxp3 expression levels with maternal acetate supplementation suggests that maternally derived acetate may influence Treg cell functional development in offspring. However, the exact mechanisms by which acetate enhances AIRE and Foxp3 expression and how they may be related, require further investigations. Collectively, these data suggest that maternally derived acetate plays important roles in shaping fetal thymic development and output. This is also consistent with the maternal-fetal correlation for serum acetate levels we found in human pregnancy, suggesting that maternal acetate likely crosses the placenta and influences fetal immunity as previously seen in a mouse model of asthma[15].

In support of our hypothesis, we found evidence of an association between low serum acetate and subsequent preeclampsia in a pregnancy cohort. Acetate levels fluctuate depending on recent dietary intake and microbial composition[50,51]. Therefore, a single measurement of serum acetate levels can only provide a snapshot and not the cumulative exposure to acetate over the duration of pregnancy. Further longitudinal studies, with larger numbers of women with preeclampsia, are required to assess acetate production in the context of microbiota composition and diet throughout preeclampsia and healthy pregnancy.

Importantly we found that the fetal immune consequences of preeclampsia are persistent, with reduced Treg frequency observed beyond infancy and into early childhood. This suggests that preeclampsia has long term implications for immune function in offspring. While the precise mechanisms underlying impaired fetal thymus development in preeclampsia remain uncertain, we suggest a key influence of maternal microbiota on this process, possibly through a direct effect of SCFAs on the fetal bone marrow and thymus.

In conclusion, this research extends our understanding of the human maternal-fetal interaction in preeclampsia and adds an alternative dimension to our conception of the developmental origins of health and disease. In addition to maternal morbidity, fetal thymic development, and Treg output are adversely affected in preeclamptic pregnancies. These associated immune changes persist into early childhood, which might explain the association between preeclampsia and later allergy in offspring[5,6], though further investigations in longitudinal cohort studies are required. Additionally, whether fetal thymic hypoplasia is mediated via dysbiosis and consequent alterations in bacterial metabolites in humans needs to be further investigated. The answers to these questions may help to establish future primary prevention strategies for preeclampsia and adverse fetal immune outcomes.

## Methods

**Human cohorts**. *Nepean cohort 1*: Prospective, case–control, comparative study: preeclamptic (n = 50) and healthy pregnant (n = 50) women were recruited. Written consent was obtained from all participants prior to entry. Preeclamptic patients were identified by daily monitoring of the antenatal ward, labor ward, and clinic attendance. Participants in the preeclampsia group underwent an ultrasound examination at Nepean Hospital (NSW, Australia) within 48 h of diagnosis. Control participants were women who presented to Nepean Hospital for a routine pregnancy ultrasound examination. Participants were recruited consecutively, subject to eligibility. Participants were eligible if they had a singleton pregnancy of gestational age ≥ 20 weeks, a normal morphology scan between 18–20 weeks and were aged between 18–40 years with no pre-existing medical conditions. Participants were excluded on the basis of chronic hypertension, illicit drug use or alcohol consumption, or obstetric complications during pregnancy (excluding preeclampsia in the preeclampsia group), or a history of intellectual or mental impairment. The Ethics Committee of the Sydney West Area Health Service, Australia approved this project (Study12/23—HREC/12/NEPEAN/45).

*Nepean cohort 2*: Prospective, (n = 887), of this group 24 developed preeclampsia. Pregnant women carrying a single fetus were recruited between 15 January 2011 and 15 July 2013, at the time of their mid-gestation ultrasound scan (17–22 weeks). In all instances, a participant information sheet was provided, and written consent was obtained by sonographers within the Perinatal Ultrasound Unit at Nepean Hospital. During the data collection period, 1155 pregnant women

were consented to participate. Fetal thymus diameter measurements were performed by one of the authors (D.P.E.) from 1139 mid-gestation scans, with an image of the right ventricular outflow tract (RVOT) unavailable in 16 scans. Once all measurements had been performed and allowing sufficient time for the last participating woman to be delivered, the ObstetriX database was searched for preeclampsia diagnoses, plus other maternal and child variables. ObstetriX is updated during and after pregnancy, by medical officers and midwives, and contributes to State-wide data collection. In total, 950 consented pregnancies were delivered at Nepean Hospital. Fifty pregnancies were excluded, based on another hypertensive diagnosis (gestational hypertension = 34, essential hypertension = 16). A further 13 pregnancies were excluded because their ultrasound scan was performed outside the range of 17–22 weeks. Participating women were not represented in more than one pregnancy. The Ethics Committee of the Sydney West Area Health Service, Australia approved this project (Study 10/14—HREC/10/NEPEAN/33).

*Nepean cohort 3*: Peripheral blood samples were obtained from healthy ($n = 66$) or preeclamptic pregnant ($n = 30$) patients. Formal consent was obtained from all patients. Preeclampsia was defined as per International Society for the Study of Hypertension in Pregnancy (ISSHP) criteria as onset of high blood pressure (>140/90) and proteinuria (>0.3 g/24 h) after 20 weeks of gestation[54]. None of the patients had HELLP syndrome or eclampsia. Exclusion criteria included gestational diabetes, infectious conditions, multiple births and chromosomal abnormalities. The Ethics Committee of the Sydney West Area Health Service, Australia approved this project (HREC 05/041).

*Nepean cohort 4*: Serum samples of paired maternal and fetal blood, collected no longer than 4 h apart, were extracted from blood samples and then stored at -80° until further analyses. Formal consent was obtained from all patients. Selection criteria were the same as in *Nepean cohort 3*. The Ethics Committee of the Sydney West Area Health Service, Australia approved this project (HREC 05/041).

*BIS cohort*: The aims and methodology of the Barwon Infant Study (BIS) have been previously described[55]. In brief, The Barwon Infant Study (BIS) is an Australian birth cohort study ($n = 1064$ mothers/1074 infants) recruited prior to 32 weeks of gestation. Exclusion criteria included birth before 32 weeks of gestation, major congenital malformation and/or genetically determined disease. Preeclampsia was defined as above. Participants with preeclampsia ($n = 31$) were compared to a randomly selected sub cohort of 293 participants without preeclampsia. All participants had provided informed consent. The study was approved by the Barwon Health Human Research and Ethics Committee (HREC 10/24).

*Fetal autopsy cohort*: Two cohorts of patients were recruited: one was from Westmead Children's Hospital (Sydney, Australia), and the other was from the Canberra Hospital (Canberra, Australia). The tissue sample were archived after biopsy, and all families of the deceased had consented to the retention of the tissues as part of the autopsy assessment. Since tissue samples belonged to Category S3 under Human Research and Ethics Committee Guidelines, which are considered as low risk, the ethics committee had approved the use of tissue samples and advised that no additional consent was required (ACT Health, Human Research and Ethics Committee, ethlr.12.182). Overall, five fetuses born to preeclamptic women in the second or third trimester of pregnancy and six age-matched controls were studied. The control patients had no clinical or biochemical evidence of preeclampsia. Mothers with other pregnancy-related complications, such as gestational diabetes or underlying autoimmune problems, were excluded from the study. Fetuses with congenital abnormalities known to affect the thymic development (e.g., 22q11 deletion) or known to be associated with any effect on T-cell populations were excluded. All cases selected histologically showed only minimal autolysis of the tissues and had their post-mortems performed close to the time of delivery. Sequential cases were selected that met the inclusion criteria. Demographics from both cohorts including gestational age, mother's age, mother's parity, the cause of fetal death, duration of preeclampsia, highest BP recorded, drug management of the blood pressure, the weight of the fetus, the weight of the thymus and spleen and placental weight and findings are summarized in Supplementary Table 5a. No information was available in relation to maternal diet or maternal or fetal serum acetate levels.

**Fetal thymus volume measurement**. Fetal thymus volume for *Nepean cohort 1* was obtained by three-dimensional ultrasound (3D-US) using a 45-degree sweep angle when a two-dimensional ultrasound (2D-US) image showed clear thymic borders. Once 3D-US image data was acquired, the information was digitally stored and later analyzed by virtual organ computer-aided analysis (VOCAL) in Viewpoint software, version 5.6.12.601 (ViewPoint Bildverarbeitung GmbH, a GE Healthcare subsidiary). Transverse, coronal, and sagittal planes were displayed by VOCAL. The transverse plane was identified and rotated in 30-degree increments for 180 degrees and the borders were manually drawn throughout the rotation. In this way, VOCAL identified the thymic borders in coronal and sagittal planes and automatically calculated the volume (Fig. 1a). Where high-quality images of the thymus could not be obtained, repeat examinations were performed. Each parameter was measured three times, with the mean of each parameter used in data analysis.

Intra-observer reliability was performed by the same examiner (S.J.) at a later date from 30 randomly selected images. To assess inter-rater reliability, 30 fetal

thymus measurements were randomly selected and repeated by an independent examiner (A.E.Q.).

**Fetal thymus diameter measurement**. Fetal thymus diameter for *Nepean cohort 2* was measured using 2D-US (Voluson 730 and E8; GE Healthcare, Zipf, Austria), with a transverse view of the fetal thymus obtained and stored using ViewPoint software, version 5.6.12.601 (ViewPoint Bildverarbeitung GmbH, a GE Healthcare subsidiary). The fetal thymus was visualized in the anterior mediastinum, within a saved image of the right ventricular outflow tract (RVOT). Fetal thymus diameter was measured as its greatest width perpendicular to a line connecting the sternum and spine (Fig. 1c).

To test the intra-observer reliability of fetal thymus diameter measures, 40 repeated measures were performed by D.P.E. several weeks after the primary measurements. Inter-rater reliability was assessed by comparing 40 primary measurements with those made by a second rater (A.E.Q.). Pregnancies were randomly selected in both instances. A good degree of correlation was observed for repeated thymus diameter measures in both intra-observer reliability (Pearson $r = 0.87$, $p < 0.001$) and inter-rater reliability (Pearson $r = 0.72$, $p < 0.001$) analyses. In addition, intra-observer and inter-rater reliability showed good correlation in both thymus diameter and volumes, with Lins concordance coefficients of 0.96 and 0.99, respectively.

**Mononuclear cell isolation**. Mononuclear cells from the Nepean Cohorts were isolated using Ficoll-Hypaque (Amersham Pharmacia, Piscataway NJ) density gradient centrifugation. All samples were frozen at −196 °C for further flow cytometric analysis. Human serum samples were obtained after centrifugation at $1000 \times g$ for 10 min, aliquoted and stored at −80 °C.

*BIS cohort*: Umbilical cord blood was collected by syringe and immediately diluted in 10 IU/mL preservative-free sodium heparin (Pfizer) in 10 mL of RPMI 1640 (Gibco, Life Technologies). Venous peripheral blood (PB) was collected at 6 months, 1 year, and 4 years of age and added to a 15 mL tube containing 10 IU/mL preservative-free sodium heparin (Pfizer). All blood was processed within 18 h of collection. Mononuclear cells were isolated by density gradient centrifugation (Lymphoprep, Axis-Shield), and flow cytometry performed in infants from preeclamptic mothers (cord blood, $n = 11$; 6-month PB, $n = 19$; 1-year-old PB, $n = 22$; 4-year-old PB, $n = 10$) and infants from non-hypertensive mothers (cord blood, $n = 434$; 6-month PB, $n = 558$; 1-year-old PB, $n = 629$; 4-year-old PB, $n = 377$).

**Flow cytometry**. *Nepean cohort 3*: Mononuclear cells were first stained using surface antibodies, after fixation and permeabilization with the Foxp3 Fix/Perm Buffer Set intracellular staining for Foxp3 was performed according to the manufacturer's instructions. Data collection was performed on a FACSVerse (BD) and data files analyzed using FlowJo software V9 (Treestar, San Carlos, CA).

*BIS cohort*: To measure Treg subsets cells were stained with antibodies to surface antibodies, and formalin fixed. After overnight fixation, cells were permeabilized (0.5% Tween) and stained with anti-Foxp3. Samples were analyzed either by the FACSCalibur (3-channel, BD) or FACSCanto II (8-channel, BD) and analyzed on complementary software. Antibodies used for flow cytometric staining of samples of the Nepean cohort 3 include: V500 anti-CD4 (BD, 562970, 1:40), APC-H7 anti-CD3 (BD, 560176, dilution 1:40), AF488 anti-Foxp3 (BioLegend, 320212, 1:50), and V450-anti-Helios (Biolegend, 137220, 1.5:100). Antibodies used for the BIS Cohort include: AF488-anti Foxp3 (BD, 560047, 1:20), PE-anti-CD4 (BD, 550630, 1:20), and PE-Cy5.5-anti-CD45RA (BD, 555490, 1:20). For both cohorts, isotype controls were used to set up the instrument for positive gating, and, once established, these settings were maintained throughout.

**Serum acetate measurement**. Maternal serum samples collected at 28 weeks of pregnancy were transported at −80 °C to the CSIRO (Commonwealth Scientific and Industrial Research Organization) laboratories, Adelaide, Australia, where serum acetate was quantified by capillary gas chromatography (GC; 5890 series II Hewlett Packard, Australia)[56]. The researchers were blinded to the experimental group.

**Nuclear magnetic resonance (NMR) spectroscopy**. Samples stored at −80 °C were thawed and 110 μL of serum transferred to a fresh 1.5 mL Eppendorf tube, and 450 μL of 1 x DPBS (Gibco Life Technologies), 50 μL of $D_2O$ (Cambridge Isotope Laboratories, Inc, MA, USA), and 10 μL of sodium 2,2-dimethyl-2-sila-pentane-5-sulfonate (DSS) (Cambridge Isotope Laboratories, Inc, MA, USA) (3.7 mg in 700 μL MilliQ water) added. Samples were centrifuged at 14,000 rpm for 10 min at 4 °C, and 550 μL transferred to an NMR tube. One-dimensional NMR spectra (128 scans) were recorded at 298 K using a standard CPMG sequence on a Bruker Avance III 600 MHz spectrometer equipped with a cryoprobe using automated data collection via IconNMR software (Bruker). Two-dimensional spectra were recorded on selected samples, including COSY, TOCSY, and $^1H$-$^{13}C$-HSQC. Samples were processed randomly. Metabolite assignments were based on the Human Metabolome Database[57] and chemical shifts derived from standard samples. For the statistical analysis, spectra were normalized to total intensity.

**Immunohistochemistry**. The spleen and thymus from the fetuses had been formalin fixed and processed into paraffin blocks using routine laboratory methods. Sequential IHC staining was performed to access the presence, location, and intensity of CD4 and Foxp3 expression of lymphocytes using the Leica Bond fully automated system (Vision Biosystems, Mt. Waverley, Australia), following a standard protocol. The cases were deidentified at time of sectioning.

Briefly, tissue sections were cut (4 μm thick) and dried in an oven at 60 °C for 1 h. Foxp3 (ab20034; Abcam, Cambridge, UK), 1:200 dilution was incubated for 15 min following 20 min heart retrieval using Bond Epitope Retrieval Solution (BERS) 2 from Leica Microsystems (Sydney, Australia), pH 8.9 to 9.1. The detection kit used was the Leica Bond Polymer Refine Detection Kit (a polymeric HRP-linker antibody conjugate system, Sydney, Australia). The same protocol was used for CD4 (NCL-L CD4–368, Novocastra 1/50 dilution). The sections were counterstained with hematoxylin to allow visualization of the nuclei, coverslipped, and viewed.

The number of CD4+ and Foxp3+ cells was counted across ten high-power fields (original magnification, x40) and scored as 0 = 0% of cells, 1 = <10%, 2 = 10–75%, 3 = >75%. The intensity of the stain was recorded as weak (1), moderate (2), or strong (3). This was performed on two separate occasions and the mean score used. The external positive control for the antibodies was an adult tonsil. Negative internal control were endothelial cells and also the omission of the primary antibody. The location of the T cells was recorded as medulla only, mainly cortex, mainly medulla and cortex only. The pathologist (J.D.) was blinded to the patient cohort at the time of assessment of the immunoreactivity of the sections. All cases were photographed and sent to a second researcher for review.

**Mouse experiments**. Germ-Free (GF) C57BL/6 mice were derived from Germ-Free Unit (Walter and Eliza Hall Institute of Medical Research). Pregnant GF mice from E18 were fed ad libitum with sodium acetate (200 mM, except where indicated) provided in the drinking water. Then 3-week-old pups (total, $n = 10$) were studied for thymic analysis. Mouse thymic analyses were performed from two or three separated cohorts of pups to obtain reproducibility. All experimental procedures involving mice had complied with all relevant ethical regulations for animal testing and research, and carried out according to protocols approved by the relevant Animal Ethics Committee of Monash University, Melbourne, Australia.

Mouse thymuses were isolated and processed for T-cell analysis or thymic epithelial cell analysis[58]. CD4+ T cells were stained using anti-mouse CD4 (Clone: RM4–5, Pacific Blue, BD Biosciences, 558107, 1:400 dilution), anti-mouse CD25 (Clone: PC61, PE Cy7, 552880, BD Biosciences, 1:400 dilution) and anti-mouse Foxp3 (Clone: FJK-16s, FITC, eBioscience, 11–5773–82, 1:200 dilution). The following antibodies for thymic epithelial cell analysis by flow cytometry or immunofluorescence studies were used: CD45 (30-F11, 1:400, BD Biosciences 557659), CD326/EpCAM (G8.8, 1:3000, eBioscience 25–5791–80), UEA-1 lectin (1:4000, Vector Labs B-1065), MHC Class II (1:5000, Biolegend 107620), Ly51 (1:2000, BD Biosciences 553735)), and AIRE (5G12, 1:100, eBioscience 3–5934–82), β5t (1:400, MBL PD021), and Keratin 14 (1:400, Biolegend 906004). For intracellular staining, cells were fixed using the Foxp3/Transcription Factor Staining Buffer Kit (eBiosciences) according to the manufacturer's instructions. The stained mouse samples were analyzed using BD LSRII flow cytometers with FACSDiva software (BD Biosciences) and FlowJo software version 9.3.2 (Tomy Digital Biology).

**Statistics**. For comparisons among two or more groups, unpaired t-test or one-way ANOVA with different types of corrections were performed for parametric data; whereas Mann–Whitney test or Kruskal–Wallis test with Dunn's correction were used for nonparametric data. Chi-square tests were used to compare groups on categorical variables. To test for correlations, Pearson's correlations coefficient was performed. All statistical analyses were performed using GraphPad Prism7 for Windows (GraphPad Prism, San Diego, CA), SAS v9.3 (SAS Institute Inc., Cary, NC, USA) or Stata 15.1 (StataCorp, College Station, TX, USA). For all analyses, a p-value of <0.05 was considered statistically significant. Apart from the above-mentioned statistical analyses, special analyses for each cohort are listed in detail as follows:

For *Nepean cohort 1*, an initial power calculation revealed a sample of 100 subjects (50 cases with preeclampsia and 50 controls) to provide over 80% power to detect an effect size of 0.6 on thymus volume at the 5% level of significance. An effect of this magnitude represented a conservative interpretation of the results of our previous study[30], which found a difference in thymus diameter corresponding to an effect size of 0.8 and obtained similar results using an approximated thymus volume based on the diameter measures. Intra-observer and inter-rater reliability were assessed used a Bland-Altman plot and a paired t-test for each assessment. In addition, the Lins Concordance Correlation coefficient was calculated to demonstrate reproducibility[59].

For *Nepean cohort 2*, a power calculation predicted a study of $n = 800$ would have at least 80% power at the two-sided 5% level of statistical significance to identify, using logistic regression modeling, an odds ratio of 0.8 for preeclampsia for every 1 mm increase in thymus diameter, given a standard deviation in thymus measurements of 2.7 mm, and assuming the underlying prevalence of preeclampsia is 2.8%. The sample calculation was performed using the

Shieh-Brien method implemented in SAS v9.2 and confirmed via simulation[60]. The estimated prevalence of preeclampsia at Nepean Hospital was based on our earlier retrospective study. The estimates of the standard deviation in thymus diameter measurements and the plausible odds ratio are also informed by the results of our previous retrospective study[30]. The effect of fetal thymus diameter on the odds of preeclampsia was evaluated in both univariate and multivariate logistic regression analyses (Model 1 and Model 2). Model 1 examined the relationship between fetal thymus diameter and preeclampsia, adjusted for gestation and maternal Body mass index (BMI). Multivariate Model 2 was similar, except adjustment was made for fetal head circumference and maternal BMI. The association between thymus diameter and gestational age was modeled using multiple linear regression methods with preeclampsia status included as an explanatory factor.

*For BIS cohort*: In examining the relationship between maternal serum acetate and subsequent preeclampsia, potential confounding variables were determined from a causal model represented by a directed acyclic graph (Supplementary Fig. 7). Testable implications were checked in R using code generated by Dagitty. Risk ratios for preeclampsia were calculated using binomial regression with a logarithmic link function, with inverse sampling probability weighting to account for the case-cohort design. Mediation was assessed using the Stata procedure "medeff". To examine potential sustained differences in nTreg proportions (birth, 6 months, and 4 years), a generalized estimating equation (GEE) regression model for longitudinal measures was used.

For the *Fetal autopsy cohort*, fetal thymic and splenic expression of CD4+ and Foxp3+ were compared between groups using an exact interpretation of the Cochrane Armitage test for trend (SAS v9.3; SAS Institute Inc., Cary, NC, USA).

**Reporting summary**. Further information on research design is available in the Nature Research Reporting Summary linked to this article.

## Data availability

All data generated or analyzed during this study are included in this published article (and its supplementary information files) or from the corresponding author upon reasonable request. The source data underlying Figs. 1b, 1d, 2b, 2d, 3, 5a–e, 6a–g and 7d, e are provided as a Source Data file.

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

## Acknowledgements

We would like to thank Dr. Tobin Northfield for assistance with the data processing. Appreciation is also extended to participants of the Barwon Infant Study and the participants of the Nepean Cohort 1–4. We would also like to thank Mrs Amanda Bullman Chief Scientist in Charge of Immunohistochemistry, Anatomical Pathology, ACT Pathology, Canberra Hospital who performed the immunohistochemical studies. We also thank Susan Abuckle and Nicky Graf for providing thymus and spleen specimens from the Children's Hospital Westmead. Drs Michael Conlon and David Topping, Commonwealth Scientific, and Industrial Research Organization, Adelaide, who measured the serum acetate levels in the BIS cohort. We are grateful to the Australian Women and Children's Research Foundation and the Nepean Medical Research Fund for supporting this project. The James Cook University NMR facility was partially funded by the Australian Research Council (LE120100015, LE160100218).

## Author contributions

M.H.: specimen collection, flow cytometry, data analyses and interpretation, and manuscript preparation; D.E.: ultrasound investigations, data analysis and interpretation, and manuscript preparation; P.H.: flow cytometry, data interpretation and manuscript preparation; E.M.: germ-free and maternal/SCFAs experimental design, data interpretation and manuscript preparation; A.C.: conceptual development and experimental design of mouse BM, thymus and AIRE study, data interpretation and manuscript preparation; B.S.N.: flow cytometry, data interpretation and manuscript preparation; J.L.R. and Y.Y.: mouse thymus and BM flow cytometry experiments, data analysis and interpretation; K.W.: performed mouse flow cytometry and immunofluorescence studies; F.C.: performed the experiments and data analyses for BIS cohort; A.Q., S.J., M.P. and R.B.: recruitment for ultrasound studies and data analyses; L.M.: data interpretation and manuscript preparation; D.W.: acetate measurements and manuscript preparation; A.P.: helped with the BIS cohort experiments; M.L.K. and M.O'H.: data interpretation and manuscript preparation; N.L.D.: planned and performed acetate measurements and data interpretation; C.R.M.: concept development of animal studies, data interpretation and

manuscript preparation; J.D.: thymus immunohistochemistry, data analysis and interpretation and manuscript preparation; The BIS Investigator Group was involved in establishing the BIS cohort; P.V. and R.N.: overall study concept, planned experiments, analyzed data, and manuscript preparation. R.N.: conceptualized the principle idea of this project.

## Additional information

**Competing interests:** The authors declare no competing interests.

## The BIS Investigator Group

Richard Saffery[9], Katrina J. Allen[9], Sarath Ranganathan[9], David Burgner[9], Leonard C. Harrison[21], Peter Sly[22] & Terry Dwyer[23]

[21]Walter and Eliza Hall Institute, Parkville, Melbourne 3052 VIC, Australia. [22]Children's Health and Environment Program, University of Queensland, Brisbane 4072 QLD, Australia. [23]The George Institute for Global Health, Newtown 2042 NSW, Australia

