## [Peer Review File · Nature Communications]

Reviewers' comments:

Reviewer #1 (Remarks to the Author):

Hu and coauthors describe a dysregulated T cell immunity in women with preeclampsia, the interplay between maternal and cord-blood immunity and how pregnancy preeclampsia also associates with long term immune development of the child and they take some of the observations further into a mouse model. The study is well performed and nicely written. It exemplifies a massive and robust workload and I would definitely support the manuscript for publication if some issues are dealt with and some additional analyses performed. The study utilizes a patchwork of many different cohorts to illustrate their research question. This is overall very nice, but also comes with limitations as they may not be directly comparable, which the authors need to acknowledge. The conclusions regarding preeclampsia and fetal thymus measurements are convincing and also between maternal and childhood T-cell populations. However moving to maternal gut composition and preeclampsia and the evidence from the mouse study performed are in my opinion not adequate to claim a direct link and this part of the manuscript should be analyzed/discussed further.

The link when going to the mouse model is not clear. I would suggest moving the BIS observations up before the mouse work, thereby a natural flow exist on why you would do the model and for your choice of acetate. However the link between maternal microbiome and preeclampsia is not well documented in the study as it is currently presented. This can perhaps be done with the BIS data, by adding more SCFA's and show why especially acetate is of importance. Alternatively; if BIS has samples on maternal microbiome these could be utilized to search for potential associations with preeclampsia. This relation is crucial to establish before postulating that it exists and before going to a mouse model of this.

Why did the authors not show whether acetate levels in the Nepean cohort 4 were also associated with preeclampsia in the women?

The mice work is interesting and thorough when analyzing thymus weight and T-cell populations, but also comes with limitations. A germ free mouse will definitely have a hampered immunity compared to a mouse with microbes. A negative control (other than keeping the mice germ free) would have been nice in the acetate study. Would e.g. butyrate supplementation induce the same or no effects? Did the authors collect microbial samples from the animals? The germ-free vs the germ-free + acetate should look the same regarding gut microbiota. This would be of interest to show that it was not just microbes introduced by accident that were responsible for the effects.

No baseline cohort data are presented for the BIS cohort, this should be included as it has been done for the other cohorts. Please add the number of preeclamptic and non-preeclamptic BIS women examined in the acetate part.

The fetal autopsy cohort adds interesting information, however, an outlier exists in both groups (PE1 and n-PE3) consider removing these – at least for robustness analyses as the values for these are much different e.g. much higher GA and birth weight.

The longitudinal data showing lower Treg populations from birth and continuing to age 4 years is a strong finding. Are the differences significant at age 4 year cross-sectionally? Please add N to the results. How was repeated measurements of the same child handled statistically?

Why did the authors only include sub-analyses on maternal steroid treatment in the Nepean cohort 3 cord-blood results only? If this observation is true, and steroids "rescue" the Treg populations, this should be evaluated for all the other analyses as well e.g. maternal blood cell populations and thymus size if the data is available.

In fig. 2a I would make the regression lines longer to illustrate, where they cross the y-axis. Please consider doing an interaction analysis for preeclampsia with the results from this figure, it is probably not significant given the lines.

Is it possible to perform a preeclampsia prediction model based on thymus size at mid-pregnancy scans? That would be extremely clinically relevant.

Thank you for letting me review this nice manuscript

Reviewer #2 (Remarks to the Author):

The authors report that fetal thymic volume and Treg numbers are decreased in preeclampsia. Treg deficiency persisted even at 4 years of age. They also report acetate normalized the decreased thymic Tregs in GF mice. Then, they observed decreased acetate levels in the maternal blood in preeclampsia patients. Despite some additional observation such as histology and flow cytometry data, decreased Tregs and fetal thymic development in eclampsia are well established in the field. Thus, low innovation is the issue with this manuscript. Regarding short-chain fatty acid connection with preeclampsia, this part is not developed enough to connect microbiota, short-chain fatty acids and eclampsia. The work is largely descriptive. More rigorous mechanistic studies should be performed to establish what the authors claim.

1. Figure 1. Impaired fetal thymic development in preeclampsia. This has been reported by others (Impaired fetal thymic growth precedes clinical preeclampsia: a case-control study. *J Reprod Immunol.* 2012 Jun;94(2):183-9)

2. Fig. 2. Reduced fetal Treg cells and fetal Treg thymic output in preeclampsia. Treg decrease in many tissues of eclampsia subjects has been reported previously by many groups.

3. Fig. 3. The longitudinal naive Treg cell changes in children from preeclamptic mothers. The naïve Treg gate is somewhat too generous including CD45RA-dim cells. Additional markers should be used for naive Tregs.

4. Fig. 4. Fetal thymus histopathology. One tissue is shown for each staining. Graphs showing many different samples from healthy and PE patients should be shown.

5. Fig. 5. Thymic defects associated with absence of gut microbiota. This figure title is not good. This is already known. What the authors want to say is acetate increases Tregs? While Treg numbers were increased. All CD4+ T cell numbers were increased regardless of whether they are Tregs or non-Tregs. The authors should examine Tregs and effector T cells also in peripheral lymphoid tissues.

Fig. 6. Serum acetate in healthy pregnancy and preeclampsia. This seems to be the major data of this manuscript but just showing a very slight difference in acetate levels (a few % differences). A key question is if this difference can really cause preeclampsia?

Dear Editor,

Thank you for giving us the opportunity to respond to the reviewer's comments:

Reviewer #1 (Remarks to the Author):

1. "...The link when going to the mouse model is not clear. I would suggest moving the BIS observations up before the mouse work, thereby a natural flow exist on why you would do the model and for your choice of acetate."

We agree with the reviewer's comments and have restructured the manuscript accordingly. "Serum acetate in healthy pregnancy and preeclampsia" section has now been moved ahead of the mouse work (see Pg. 11).

2. "... the link between maternal microbiome and preeclampsia is not well documented in the study as it is currently presented. This can perhaps be done with the BIS data, by adding more SCFA's and show why especially acetate is of importance. Alternatively; if BIS has samples on maternal microbiome these could be utilized to search for potential associations with preeclampsia. This relation is crucial to establish before postulating that it exists and before going to a mouse model of this".

We have now analysed the other SCFAs, namely butyrate and propionate, in the BIS cohort sample set. We found that, along with acetate, maternal butyrate and propionate levels were also associated with subsequent PE; however, serum acetate levels in pregnant women were at least 10-fold higher than butyrate and propionate levels. This data has now been included in Figure 5a to 5d of the manuscript (see below Figure) and the Result Section (Pg. 11, Ln. 205-208: *Consistent with previous findings, serum acetate levels were at least 10-fold higher compared to butyrate and propionate (Fig. 5a). In addition, on weight logarithmic regression analysis, there was also evidence of associations between lower serum SCFAs and subsequent preeclampsia (Fig. 5b, 5c and 5d)*). The substantially higher concentration of acetate than butyrate or propionate makes it a more likely candidate to cross the placenta in sufficient quantities to influence fetal immune development, thus providing a strong rational for using acetate in our mouse model.

Serum acetate in healthy pregnancy and preeclampsia (Fig.5a to 5d in our manuscript)

In addition, our co-authors have previously shown that high fibre diet in mice leads to significantly higher levels of SCFA, particularly acetate, and can influence fetal immune development ¹. This has now been clarified in the Background (Pg. 5 Ln 91 to 94: *Furthermore, in a murine model, supplementation of high fibre or direct supplementation of acetate significantly reduced both systolic and diastolic blood pressures* ², a finding highly relevant to preeclampsia with hypertension as a defining clinical feature) and the Discussion Section (Pg. 16, Ln 303-305: *Moreover, the fact that acetate also has a direct effect on systemic blood pressure* ², implies beyond its immune modulating effects a special role of this SCFA in preeclampsia).

Unfortunately, data directly relating the maternal microbiome to subsequent preeclampsia are unavailable in the cohorts included in this paper. It is important to recognise that there are very substantial logistical challenges to longitudinal assembly of the samples and measures required to evaluate the relationship between the pre-existing microbiome early in pregnancy to subsequent preeclampsia, and it is therefore unlikely that such data will be produced for some time.

3. Why did the authors not show whether acetate levels in the Nepean cohort 4 were also associated with preeclampsia in the women?

The timing of serum collection in Nepean was different to that of the BIS cohort and the sample size too small for a comparison (PE: n=8). In the Nepean cohort, serum acetate levels were determined at delivery from pairs of maternal and cord blood. The purpose of these measurements was mainly to show a correlation between cord and maternal blood acetate levels. The comparison of acetate levels from the BIS cohort were from serum samples taken at 28 weeks gestation to show differences in serum acetate levels between PE and non-PE pregnancies.

4. The mice work is interesting and thorough when analysing thymus weight and T-cell populations, but also comes with limitations. A germ-free mouse will definitely have a hampered immunity compared to a mouse with microbes. A negative control (other than keeping the mice germ free) would have been nice in the acetate study. Would e.g. butyrate supplementation induce the same or no effects? Did the authors collect microbial samples from the animals? The germ-free vs the germ-free + acetate should look the same regarding gut microbiota. This would be of interest to show that it was not just microbes introduced by accident that were responsible for the effects.

We do not have an alternative supplement control. We did not treat pregnant mice with butyrate in drinking water given that in previous experience mothers eat their litters. The germ-free mice at the Walter and Eliza Hall Institute are regularly monitored for contamination and there has been no indication for accidental contamination on routine monitoring of the facilities. Both the water as well as the acetate have undergone sterilisation before introducing them to the germ-free environment

5. No baseline cohort data are presented for the BIS cohort, this should be included as it has been done for the other cohorts. Please add the number of preeclamptic and non-preeclamptic BIS women examined in the acetate part.

The baseline cohort data for the BIS cohort has been included now in the supplementary Table S4 (Pg. 54). The number of PE and non-PE had been documented previously in the methods section (Pg. 21 Ln. 400-401 and Pg. 22 Ln. 434-436) as well as in the figure legend of Fig. 3 (Pg. 40, Ln. 823-825) and Fig. 5 (Pg. 43, Ln. 839-840).

6. The fetal autopsy cohort adds interesting information however; an outlier exists in both groups (PE1 and n-PE3) consider removing these – at least for robustness analyses as the values for these are much different e.g. much higher GA and birth weight.

We have followed the reviewer's advice and have omitted PE1 and n-PE3. Table S5 has been updated with new p-values (Pg. 56). Corresponding number of PE (5 instead of 6) and non-PE (6 instead of 7) have been modified in the result and method sections (Pg. 10 Ln 193 and 197; Pg. 21 Ln 406).

7. The longitudinal data showing lower Treg populations from birth and continuing to age 4 years is a strong finding. Are the differences significant at age 4-year cross-sectionally? Please add N to the results. How were repeated measurements of the same child handled statistically?

At the 4yr time point there is weak evidence for a cross-sectional difference in nTreg (-0.8%, 95% CI -1.8, 0.2, p=0.12, Two sample t-test) or (-0.92%, p=0.087, Wilcoxon rank sum).

The number of PE and non-PE had been documented previously in the methods section (Pg. 22 Ln. 434-436) as well as in the figure legend (Fig. 3, Pg. 40, Ln. 823-825): *Offspring born to preeclampsia mothers - cord blood, n=11; six-month PB, n=19; 1-year-old PB, n=22; 4-year-old PB, n=10, compared to offspring from non-hypertensive mothers - cord blood, n=434; six-month PB, n=558; 1-year-old PB, n=629; 4-year-old PB, n=377.*

A generalized estimating equation (GEE) regression model was used to handle the repeated measurements of the same child. It is now clarified in the method section on Pg 29, Ln 582-584: *“To examine potential sustained differences in nTreg proportions (birth, 6 and 12 months, and 4 years), a generalized estimating equation (GEE) regression model for longitudinal measures was used”.*

8. Why did the authors only include sub-analyses on maternal steroid treatment in the Nepean cohort 3 cord-blood results only? If this observation is true, and steroids “rescue” the Treg populations, this should be evaluated for all the other analyses as well e.g. maternal blood cell populations and thymus size if the data is available.

For thymus size, steroid treatment plays no role as measurements were made at 18-21 weeks before steroids are given. As for the BIS cohort, only 3 out of 324 pregnant women had

received steroid treatment, therefore, it is impossible to evaluate the effect of steroids (refer to the participants characteristics in Table S4, Page 55).

9. In Fig. 2a I would make the regression Ln. longer to illustrate, where they cross the y-axis. Please consider doing an interaction analysis for preeclampsia with the results from this figure, it is probably not significant given the Ln.

We have extended the regression line to cross the y-axis as suggested. Interaction analysis has also been performed as requested. When doing multiple regression in SPSS, we can create a regression model: $CB = 2.880 + 0.506 \times MB - 0.791 \times PE$ (where PE=1 for PE and 0 for non-PE). Both factors contribute significantly to this equation (see below).

Model Summary

Model	R	R Square	Adjusted R Square	Std. Error of the Estimate
1	.620 ^a	.385	.370	1.37305

a. Predictors: (Constant), PE, MB

Coefficients^a

Model		Unstandardized Coefficients		Standardized Coefficients	t	Sig.	95.0% Confidence Interval for B	
		B	Std. Error	Beta			Lower Bound	Upper Bound
1	(Constant)	2.880	.507		5.684	.000	1.873	3.888
	MB	.506	.080	.546	6.309	.000	.346	.665
	PE	-.791	.326	-.210	-2.423	.018	-1.440	-.142

a. Dependent Variable: CB

However, if the interaction between maternal blood and PE is involved, as the reviewer mentioned, it becomes insignificant (see below), indicating that the differences in slopes of the red (PE) and black (non-PE) line is non-significant.

Model Summary

Model	R	R Square	Adjusted R Square	Std. Error of the Estimate
1	.623 ^a	.388	.366	1.37776

a. Predictors: (Constant), MBxPE, MB, PE

Coefficients^a

Model		Unstandardized Coefficients		Standardized Coefficients	t	Sig.	95.0% Confidence Interval for B	
		B	Std. Error	Beta			Lower Bound	Upper Bound
1	(Constant)	2.681	.595		4.509	.000	1.499	3.863
	MB	.539	.096	.583	5.634	.000	.349	.730
	PE	-.174	1.008	-.046	-.172	.864	-2.179	1.832
	MBxPE	-.114	.177	-.170	-.647	.519	-.465	.237

a. Dependent Variable: CB

10. Is it possible to perform a preeclampsia prediction model based on thymus size at mid-pregnancy scans? That would be extremely clinically relevant.

We are grateful to the reviewer for his suggestion. Receiver Operating Characteristic (ROC) curves were generated to evaluate fetal thymus diameter at mid-gestation as a predictive tool for preeclampsia. Since our group had previously found that fetal head growth was also associated with preeclampsia after adjusting cofounders ³, the fetal head circumference was also included in the model. The best prediction model including thymic size between 17- and 21-week gestation as well as fetal head circumference at mid-gestation, gestational age and maternal BMI showed an AUC of 0.81 indicating that these parameters might be useful predictors of preeclampsia.

This result has now been included in the supplementary Fig. S1 (see below) and the Result Section (Pg. 8 Ln 146-153: *Receiver Operating Characteristic (ROC) curves were generated to evaluate fetal thymus diameter at mid-gestation as a predictive tool for preeclampsia. Fetal thymus diameter at mid-gestation alone was found to have an area under the curve (AUC) of 0.71. Together with gestation and maternal BMI, AUC increased to 0.78. Since our group had previously found that increased fetal head growth was also associated with preeclampsia after adjusting cofounders ³, we then included the fetal head circumference in the model. When fetal thymus diameter at mid-gestation, fetal head circumference at mid-gestation, gestational age and maternal BMI were included, we could reach an AUC of 0.81 (Fig. S1).*

Fig. S1 Fetal thymus diameter at mid-gestation as a predictive tool for preeclampsia.

Reviewer #2 (Remarks to the Author):

1. Despite some additional observation such as histology and flow cytometry data, decreased Tregs and fetal thymic development in eclampsia are well established in the field. Thus, low innovation is the issue with this manuscript. Regarding short-chain fatty acid connection with preeclampsia, this part is not developed enough to connect microbiota, short-chain fatty acids and eclampsia. The work is largely descriptive. More rigorous mechanistic studies should be performed to establish what the authors claim.

We are aware that our own group has shown a reduction in maternal Treg and reduced fetal thymus size in preeclampsia previously^{4,5}. In this study we extend these findings to show that the reduced Treg proportion is sustained throughout the first year and into early childhood for the first time. Furthermore, we provide novel and compelling mechanistic data to link these observations. Our work demonstrates that maternal serum acetate levels are reduced in preeclampsia and correlates with fetal acetate level. The significance of this is underpinned by data showing that supplementation of acetate in germ free mice rescues the thymic/T cell development in this model. This data is highly novel and relevant, especially in the absence of a good PE mouse model.

2. Fig. 1. Impaired fetal thymic development in preeclampsia. This has been reported by others (Impaired fetal thymic growth precedes clinical preeclampsia: a case-control study. J Reprod Immunol. 2012 Jun;94(2):183-9)

We are aware that we and others have published in this area previously. However, previous data was retrospective in nature and our current data includes a prospective cohort, where thymus measurements were undertaken between 18- and 21-week gestation, way before the onset of preeclampsia. We have further added data evaluating the utility of thymic measurement and fetal head circumference in predicting preeclampsia and showed that these parameters may be useful in PE prediction, which is explained in supplementary Fig. S1 (see above Fig. S1 in the rebuttal) and an additional paragraph on Pg. 8 Ln 146-153 (*Receiver Operating Characteristic (ROC) curves were generated to evaluate fetal thymus diameter at mid-gestation as a predictive tool for preeclampsia. Fetal thymus diameter at mid-gestation alone was found to have an area under the curve (AUC) of 0.71. Together with gestation and*

maternal BMI, AUC increased to 0.78. Since our group had previously found that increased fetal head growth was also associated with preeclampsia after adjusting cofounders ³, we then included the fetal head circumference in the model. When fetal thymus diameter at mid-gestation, fetal head circumference at mid-gestation, gestational age and maternal BMI were included, we could reach an AUC of 0.81 (Fig. S1)). This is novel and highly clinically significant.

3. Fig. 2. Reduced fetal Treg cells and fetal Treg thymic output in preeclampsia. Treg decrease in many tissues of eclampsia subjects has been reported previously by many groups. We agree that reduced MATERNAL Treg cells has been noted in the past and our group has also previously reported on this ⁴. However, few groups have studied fetal Treg cells or thymic output in preeclamptic pregnancies. This was the focus of this report and demonstrated importantly that preeclampsia does impact on fetal immune development.

4. Fig. 3. The longitudinal naïve Treg cell changes in children from preeclamptic mothers. The naïve Treg gate is somewhat too generous including CD45RA-dim cells. Additional markers should be used for naïve Tregs.

The gating and assessment of the naïve Treg population in this study has previously been published ⁶. The gating strategy is based on the seminal study by Miyara et al ⁷. In this work, and studies performed since then, the population of naïve Treg were clearly delineated by gating to the CD4⁺ T-cells and using antibodies to CD45RA and FoxP3 to differentiate the CD45RA^{high}/FoxP3⁺ population (see figure below).

Separation and delineation of three CD4⁺/CD45RA/FoxP3⁺ populations as published by Miyara et al. Population I (CD45RA^{high}/FoxP3⁺) defines the naïve Treg (Image taken from Fig. 1 of the 2009 paper) ⁷.

In the 2009 study the population phenotype was further confirmed by assessment of their DNA methylation status, their proliferative capacity and their secreted cytokine profile ⁷. We do not believe that any additional markers are required to discriminate naïve Treg. Also, our gating compares well to the Miyara protocol (see figure below) and does not seem too generous for the CD45RA-dim cells.

Separation and delineation of three CD4⁺/CD45RA/FoxP3⁺ populations (Fig.3a in our manuscript).

5. Fig. 4. Fetal thymus histopathology. One tissue is shown for each staining. Graphs showing many different samples from healthy and PE patients should be shown.

We have provided a table with the requested information (supplementary Table S5, Pg. 56) and an example of the histopathology findings in the current manuscript. It is possible to provide all the histopathology sections in the supplementary data, but it would create an immensely large file. Hence, we are not sure what displaying all histopathology sections would add to the manuscript and are reluctant to expand the manuscript with unnecessary data.

6. Fig. 5. Thymic defects associated with absence of gut microbiota. This figure title is not good. This is already known. What the authors want to say is acetate increases Tregs? While Treg numbers were increased. All CD4+ T cell numbers were increased regardless of whether they are Tregs or non-Tregs. The authors should examine Tregs and effector T cells also in peripheral lymphoid tissues.

We have changed the title (now Fig 6 of our manuscript on Pg. 44) to “Reduced Foxp3 MFI in thymic CD4SP Tregs from germ free pups recovered with acetate supplementation” and added our new findings.

Thymic defects associated with absence of gut microbiota has not been previously described to our knowledge. We maintain that this is a novel finding. The paper focused on thymic development and its dependency on acetate to confirm the findings of the human studies. Although we do not have data from other peripheral lymphoid tissues from the mouse studies, we do have now novel data in the new figures (Fig. 6, Fig. 7 and supplementary Fig. S2 in our manuscript, see below Figures, demonstrating that changes in thymus weight can be explained by increased BM (bone marrow) cellularity. Similarly, we have novel data on significant alterations in AIRE expression intrinsic to thymic epithelial cells in germ free pups, which is recovered with acetate supplementation. This may underlie the recovered Foxp3 MFI in Tregs found with acetate supplementation and a trend towards improved Treg numbers.

Reduced Foxp3 MFI in thymic CD4SP Tregs from germ free pups recovered with acetate supplementation. (Fig. 6 in our manuscript).

Reduced thymus expression levels of the autoimmune regulator (AIRE) protein in germ free pups recovered with acetate supplementation. (Fig. 7 in our manuscript).

The Role of the Thymus Architecture on Thymus Expression Levels of the Autoimmune Regulator (AIRE) Protein. (supplementary Fig. S2 in our manuscript).

To explain our findings, three major sub-sections have been added to the Results (from Pg. 12 Ln. 229 to Pg. 14 Ln. 274, see below), and a few sentences have been added to the Discussion on Pg. 16-17, Ln. 307-323 (*In a mouse model, we showed that absence of a maternal gut microbiota leads to offspring with reduced BM derived LMPP precursors that may underpin the substantial reduction in thymic size and thymic output. This could be rescued by supplementing maternal drinking water with acetate, which is the most abundant SCFA in human serum*⁸⁻¹⁰. *Decreased Treg cell Foxp3 levels were also found coincident with a reduction in AIRE expression in specialized thymic epithelial cells in the absence of a maternal gut microbiota. Both were rescued with maternal acetate supplementation. Treg cell generation is influenced by levels of Aire expression in medullary thymic epithelial cells*¹¹ *and their functional status is reflected by the levels of Foxp3*¹². *While the majority of Aire expression is found deep within the thymic medulla, we identified a small proportion of Aire expressing mature cortical thymic epithelial cells located at the cortico-medullary junction that were influenced by maternal gut microbiota. Our findings that a reduction in this specialised Aire⁺ epithelial cell population influenced the levels of Foxp3 expression in Treg cells, suggests a role in Treg development. While Aire expressing medullary epithelial cells express specialised tissue restricted self-antigens, these cortical epithelial cells may present abundant, ubiquitous self antigens early in Treg development*¹³. *Thus, coincident recovery of Aire expression in cortical epithelial cells and increased Foxp3 expression levels, suggests maternally derived acetate may also influence Treg cell functional development in offspring).*

Impaired T cell development in germ free mice can be rescued by maternal acetate consumption

Mounting evidence suggests that SCFAs and other bacterial metabolites influence T cell/Treg cell development^{1,14-17}. *We therefore investigated whether maternal bacterial load could*

affect fetal thymic development by comparing thymus weight of pups (21 days old) born to germ-free (GF) mice with specific pathogen free (SPF) mice. Thymic weight (**Fig. 6a**) and thymocyte cellularity (**Fig. 6b**) in GF mice was approximately half that of SPF mice, which mirrored the findings in preeclampsia. Since our human cohort showed that serum acetate levels were 10-fold higher compared to butyrate and propionate, we then supplemented the GF pregnant mice with acetate in the drinking water and analysed pups. Interestingly, maternal acetate supplementation rescued both thymic weight and thymocyte cellularity in pups to levels comparable to SPF mice (**Fig. 6a, 6b**). This suggests that acetate produced by bacteria in maternal SPF mice contributed to fetal thymic development. We also found bone marrow (BM) cellularity was significantly decreased in GF mice compared to SPF controls (**Fig. 6c**); in particular, lymphoid-primed multipotent progenitors ($CD62L^+$ LMPP; **Fig. 6d**) which supply the thymus with progenitor cells for T cell development. Maternal acetate supplementation improved $CD62L^+$ LMPP cell proportions in pups, which may have contributed to alleviation of thymic defects.

Fetal $CD4^+$ T cells and Treg development is potentiated by maternal acetate consumption

We then compared the development of T cell subsets in GF and SPF fetuses. Offspring of GF mice had decreased $CD4^+$ T cell numbers (**Fig. 6e**) and fewer $CD4^+$ $Foxp3^+$ Tregs (**Fig. 6f**). In addition, reduced abundance of *Foxp3* protein per cell, measured by mean fluorescence intensity (MFI), was evident in GF mice (**Fig. 6g**). Interestingly, maternal acetate supplementation restored the total number of $CD4^+$ T cells in pups to that of SPF mice. While Treg numbers did not increase, acetate supplementation rescued *Foxp3* MFI in Treg cells (**Fig. 6g**), with implications in Treg function. Together these results suggest that bacteria-derived acetate is important for fetal $CD4^+$ T cell and Treg cell development and dietary acetate supplementation can correct defects resulting from an absent microbiota.

SCFA acetate supplementation recovers Treg development through changes in thymus expression levels of the AIRE protein

AIRE is important for Treg cell generation early in life¹⁸ and *Foxp3* levels are important for Treg function¹², we therefore investigated for alterations in *Aire* expression in thymic epithelial cells subsets that could explain the variation in Treg *Foxp3* expression with maternal diet. Using europaeus agglutinin 1 (UEA1) lectin and major histocompatibility complex II (MHC II) expression, $EpCAM^+$ thymic epithelial cells (TEC) can be broadly

divided into cortical (cTEC; UEAI⁻) and medullary (mTEC; UEAI⁺) subsets by flow cytometry analysis (Fig 7a). The majority of Aire⁺ cells are found in the keratin 14⁺ medulla region (Supplementary Fig S2). However, we also found a minority of Aire⁺ cells with a cortical phenotype ($\beta 5t^+$, keratin 14⁻) situated at the cortico-medullary junction (white arrows, Supplementary Fig S2), that were influenced by maternal microbiota. While this Aire⁺ cortical epithelial cell population is not well described, likely due to technical complexities, their association with Foxp3 expression suggests a role in AIRE-dependent Treg development. Representative FACS plots demonstrate a loss of AIRE expression in the cTEC subset of GF mice compared to SPF controls, and this is recovered with acetate treatment (Fig 7b). These changes demonstrate both proportional and numerical significance (Fig. 7c). No significant alterations in AIRE⁺ mTECs were observed (Fig 7d, e). Therefore, the recovered numbers of AIRE⁺ cTECs with acetate treatment may underpin the improved Foxp3 MFI in Tregs.

7. Fig. 6. Serum acetate in healthy pregnancy and preeclampsia. This seems to be the major data of this manuscript but just showing a very slight difference in acetate levels (a few % differences). A key question is if this difference can really cause preeclampsia?

We have now provided comparative data of acetate, butyrate and propionate and showed that all these SCFAs are significantly decreased in PE (Fig. 5a to 5d, refer to Fig. 1 in the rebuttal). These differences were detected at 28 weeks of gestation. It is very likely that reduced acetate levels in PE might be more pronounced prior to 28 weeks causing thymic size reduction as described between 17-21 weeks. Unfortunately, we do not have acetate levels from the first half of pregnancy. We do not claim that reduced acetate level is the sole cause of PE, but our data does provide evidence that it may contribute to PE development and is likely responsible for the impaired fetal thymic development seen in PE pregnancies.

References

- 1 Thorburn, A. N. et al. Evidence that asthma is a developmental origin disease influenced by maternal diet and bacterial metabolites. Nature communications 6, 7320, doi:10.1038/ncomms8320 (2015).

- 2 Marques, F. Z. et al. >High-Fiber Diet and Acetate Supplementation Change the Gut Microbiota and Prevent the Development of Hypertension and Heart Failure in Hypertensive MiceClinical Perspective. *Circulation* 135, 964-977, doi:10.1161/circulationaha.116.024545 (2017).
- 3 Eviston, D. P., Minasyan, A., Mann, K. P., Peek, M. J. & Nanan, R. K. Altered Fetal Head Growth in Preeclampsia: A Retrospective Cohort Proof-Of-Concept Study. *Frontiers in pediatrics* 3, 83, doi:10.3389/fped.2015.00083 (2015).
- 4 Santner-Nanan, B. et al. Systemic increase in the ratio between Foxp3+ and IL-17-producing CD4+ T cells in healthy pregnancy but not in preeclampsia. *Journal of immunology* 183, 7023-7030, doi:10.4049/jimmunol.0901154 (2009).
- 5 Eviston, D. P. et al. Impaired fetal thymic growth precedes clinical preeclampsia: a case-control study. *Journal of reproductive immunology* 94, 183-189, doi:10.1016/j.jri.2012.04.001 (2012).
- 6 Collier, F. M. et al. The ontogeny of naive and regulatory CD4(+) T-cell subsets during the first postnatal year: a cohort study. *Clinical & translational immunology* 4, e34, doi:10.1038/cti.2015.2 (2015).
- 7 Miyara, M. et al. Functional delineation and differentiation dynamics of human CD4+ T cells expressing the FoxP3 transcription factor. *Immunity* 30, 899-911, doi:10.1016/j.immuni.2009.03.019 (2009).
- 8 Muir, J. G. et al. Resistant starch in the diet increases breath hydrogen and serum acetate in human subjects. *The American journal of clinical nutrition* 61, 792-799 (1995).
- 9 Cummings, J. H., Pomare, E. W., Branch, W. J., Naylor, C. P. & Macfarlane, G. T. Short chain fatty acids in human large intestine, portal, hepatic and venous blood. *Gut* 28, 1221-1227 (1987).

- 10 Wolever, T. M., Josse, R. G., Leiter, L. A. & Chiasson, J. L. Time of day and glucose tolerance status affect serum short-chain fatty acid concentrations in humans. *Metabolism: clinical and experimental* 46, 805-811 (1997).
- 11 Lin, J. et al. Increased generation of Foxp3(+) regulatory T cells by manipulating antigen presentation in the thymus. *Nature communications* 7, 10562, doi:10.1038/ncomms10562 (2016).
- 12 Chauhan, S. K., Saban, D. R., Lee, H. K. & Dana, R. Levels of Foxp3 in regulatory T cells reflect their functional status in transplantation. *Journal of immunology* 182, 148-153 (2009).
- 13 Nishijima, H. et al. Ectopic Aire Expression in the Thymic Cortex Reveals Inherent Properties of Aire as a Tolerogenic Factor within the Medulla. *Journal of immunology* 195, 4641-4649, doi:10.4049/jimmunol.1501026 (2015).
- 14 Smith, P. M. et al. The microbial metabolites, short-chain fatty acids, regulate colonic Treg cell homeostasis. *Science (New York, N.Y.)* 341, 569-573, doi:10.1126/science.1241165 (2013).
- 15 Arpaia, N. et al. Metabolites produced by commensal bacteria promote peripheral regulatory T-cell generation. *Nature* 504, 451-455, doi:10.1038/nature12726 (2013).
- 16 Furusawa, Y. et al. Commensal microbe-derived butyrate induces the differentiation of colonic regulatory T cells. *Nature* 504, 446-450, doi:10.1038/nature12721 (2013).
- 17 Marino, E. et al. Gut microbial metabolites limit the frequency of autoimmune T cells and protect against type 1 diabetes. *Nature immunology* 18, 552-562, doi:10.1038/ni.3713 (2017).
- 18 Yang, S., Fujikado, N., Kolodin, D., Benoist, C. & Mathis, D. Immune tolerance. Regulatory T cells generated early in life play a distinct role in maintaining self-tolerance. *Science* 348, 589-594, doi:10.1126/science.aaa7017 (2015).

Reviewers' comments:

Reviewer #1 (Remarks to the Author):

Thank you for letting me review this revision of the manuscript. All my previous raised concerns have been addressed sufficiently by the authors in their responses. I really like the addition of the nice prediction model based on the scanning data.

Of curiosity, there is evidence (as the authors also report here) of a higher risk of preeclampsia if the mother is nulliparous. Can part of the mechanism behind this association be explained by the author's findings in the current manuscript or is this something completely different?

Best regards,
Jakob Stokholm

Reviewer #2 (Remarks to the Author):

The authors revised the manuscript by providing some more information. However, the manuscript still has many weaknesses. More specifically, it is not convincingly shown if acetate produced by microbiota really supports thymus development. It is also not clear it is just Tregs that are supported by the SCFAs. Moreover, there is no data showing the mechanism of AIRE regulation by SCFAs. In methods, the authors used an extremely high concentration of SCFAs to increase SCFA levels in mice, which is supraphysiological and can cause artificial changes in pregnant mice.

1. Introduction,

"we describe a mechanistic link between maternal gut microbial metabolites and fetal immune development in preeclampsia."

Unlike the claim, there is no clear mechanistic data explaining how acetate increased Tregs in eclampsia. More specially, no data are shown to provide mechanisms for increased Tregs and AIRE expression by acetate supplementation. What authors show are simple association of smaller thymus size with preeclampsia.

2. Fetal thymus size is smaller in preeclampsia. This should affect both Tregs and non-Tregs but the authors describe only about Tregs. This is oversimplification of the observed phenomenon.

3. Fig. 1. Impaired fetal thymic development in preeclampsia. Decreased thymus development may be due to smaller body size in preeclampsia. The growth of fetus should be also shown to show the defective thymic development is unique in the fetus.

3. Fig. 2. "%Foxp3+ (Mean) within CD4+ cells in blood" are shown in many figures. Thymus produce more non-Tregs than Tregs. Total T cell output should be shown rather than showing only Treg frequency among CD4+ T cells. The frequency may not have to change if total T cell output is down in preeclampsia. The authors should show data for conventional CD4 and CD8 T cell subsets.

4. Fig. 3. The simple gating based on CD45RA and FoxP3 is not a reliable way to identify Treg subsets due to rather faint staining of FoxP3. This is prone to include non-Tregs. Utilization of other antigens such as CD25, CTLA4 and others should help with the identification.

5. Fig. 4. For rigor and reproducibility of the data, the size of the thymic lobules and other important features should be plotted in graphs.

6. Fig.5. Not clear why the authors are showing the SCFA concentration data after log-transformed. As it presented, the differences are not clear.

7. There is no evidence presented that acetate normalize Treg numbers through AIRE regulation. What is the mechanism for AIRE gene regulation by acetate?

8. Figure 6 shows that CD4 SP T cell numbers are also decreased in GF mice. This indicates that it is not just Tregs that are affected. Despite this, the authors emphasize the changes in Tregs only.

9. In methods, they fed mice with extremely high concentration of sodium acetate (200 mM). This is physiologically irrelevant. In mice, the acetate level in the gut lumen is much lower than this. How much is required to change SCFA levels in fetal blood and thymus architecture?

Dear Editor,

Thank you for giving us the opportunity to respond to the reviewer's comments:

Reviewer #1

Of curiosity, there is evidence (as the authors also report here) of a higher risk of preeclampsia if the mother is nulliparous. Can part of the mechanism behind this association be explained by the author's findings in the current manuscript or is this something completely different?

This is an interesting question, which cannot be explained by our findings presented in the current manuscript. This particular feature of preeclampsia might involve maternal Treg memory from first pregnancy to confer protection against preeclampsia in consecutive pregnancies.

Reviewer #2 (Remarks to the Author):

The authors revised the manuscript by providing some more information. However, the manuscript still has many weaknesses. More specifically, it is not convincingly shown if acetate produced by microbiota really supports thymus development. It is also not clear it is just Tregs that are supported by the SCFAs. Moreover, there is no data showing the mechanism of AIRE regulation by SCFAs. In methods, the authors used an extremely high concentration of SCFAs to increase SCFA levels in mice, which is supraphysiological and can cause artificial changes in pregnant mice.

A point by point response to the issues raised by the referee has been provided below. We have also toned down our claims within the manuscript as advised by the reviewer and the editor.

1. *Introduction, "we describe a mechanistic link between maternal gut microbial metabolites and fetal immune development in preeclampsia." Unlike the claim, there is no clear mechanistic data explaining how acetate increased Tregs in eclampsia. More specially, no data are shown to provide mechanisms for increased Tregs and AIRE expression by acetate supplementation. What authors show are simple association of smaller thymus size with preeclampsia. Fetal thymus size is smaller in preeclampsia. This*

should affect both Tregs and non-Tregs but the authors describe only about Tregs. This is oversimplification of the observed phenomenon.

We have now changed this section in the introduction as recommended by the referee (page 6, lines 106-109): *“In this study, we describe an association between maternal gut microbial metabolites and fetal immune development in preeclampsia. Specifically, we demonstrate a link between acetate and fetal thymic development and output, with striking differences between non-preeclamptic and preeclamptic pregnancies.”*

Abstract (page 3, line 58-59): *These findings suggest a potential role of acetate in the pathogenesis of preeclampsia, and for immune events in offspring.*

Discussion (page 15, line 278-279): *These studies provide a link between maternal gut bacterial metabolite acetate, thymus and Treg development in the context of preeclampsia.*

We agree with the reviewer that treatment with acetate correlates with increased expression of AIRE, which does not necessarily mean that acetate directly promotes AIRE expression. The regulation of AIRE expression is poorly characterised in the current literature. Delineation of the mechanisms behind the effects of acetate supplementation on AIRE expression is beyond the scope of this manuscript. The sentence *“However, the exact mechanisms by which acetate enhances Aire and Foxp3 expression and how they may be related, require further investigations.”* has been now added in Discussion on page 17, lines 331-333.

We agree that reduced fetal thymus size could possibly affect both Tregs and non-Tregs; however, in our study, we only found changes in Tregs. This point is addressed in more detail in response to question 3.

2. Fig. 1. Impaired fetal thymic development in preeclampsia. Decreased thymus development may be due to smaller body size in preeclampsia. The growth of fetus should be also shown to show the defective thymic development is unique in the fetus.

We agree that fetal thymus size is related to fetal size. To address this, fetal thymus measurements were adjusted for fetal size parameters including estimated fetal weight (Nepean Cohort 1) and fetal head circumference (Nepean Cohort 2). In terms of Fig. 1, we believe that the addition of fetal growth data would unnecessarily clutter the graphs.

3. *Fig. 2. “%Foxp3+ (Mean) within CD4+ cells in blood” are shown in many figures. Thymus produce more non-Tregs than Tregs. Total T cell output should be shown rather than showing only Treg frequency among CD4+ T cells. The frequency may not have to change if total T cell output is down in preeclampsia. The authors should show data for conventional CD4 and CD8 T cell subsets.*

We found no evidence that preeclamptic pregnancies were associated with decreased percentages of thymic T cells (CD3 T cells), or conventional CD4/CD8 T cell subsets (see below figure) in contrast to the changes found within the Treg cells compartment. The sentence “*In contrast, we found no evidence that preeclamptic pregnancies were associated with changes in percentages of fetal thymic T cells (CD3 T cells), or conventional CD4/CD8 T cell subsets (data not shown).*” has now been added into Discussion on page 15, lines 294-296.

There is compelling data suggesting that changes in percentages of Treg cells within CD4⁺ cells play a central role in many immune-mediated pathologies¹⁻⁴. In line with this, our own group has shown reduced percentages in maternal Tregs to be a feature of altered immunity in preeclampsia⁵⁻⁷. Hence, differences in percentages of offspring Treg cells described in this study between the preeclampsia and non-preeclampsia groups seems highly relevant.

We have no data on the absolute (total) numbers of thymic Treg cells or CD4/CD8 T cell subsets. To calculate total numbers of these subsets would have required the availability of full blood counts, which were not systematically collected for any of the cohorts.

4. *Fig. 3. The simple gating based on CD45RA and FoxP3 is not a reliable way to identify Treg subsets due to rather faint staining of FoxP3. This is prone to include non-Tregs. Utilization of other antigens such as CD25, CTLA4 and others should help with the identification.*

Our approach was based on the seminal studies of Shimon Sakaguchi⁸⁻¹⁰ who showed that “the combination of FoxP3 and CD45RA expression can dissect FoxP3⁺ cells into three subpopulations: (i) CD45RA⁺FoxP3^{low} naive Tregs (nTregs); (ii) CD45RAFoxP3^{high} effector Tregs (eTregs), both of which are potently suppressive in vitro; and (iii) non-suppressive, cytokine secreting CD45RA⁺FoxP3^{low} non-Tregs”¹⁰.

Sakaguchi et al also state that “As the expression levels of FoxP3 are proportional to those of CD25, the use of CD25 and CD45RA enables the isolation of the three populations as live cells”¹⁰. In other words, CD25 can be used as a surface marker for isolation of individual populations of Tregs, but is not necessary in combination with FoxP3 for the assessment and discrimination of nTreg.

As shown in Figure 3 the degree of FoxP3 staining was always strong enough to discriminate FoxP3-positive from FoxP3-negative cells. Finally, we have published previously on the nTreg measured in this way¹¹⁻¹³, and there are many other publications that have used the same strategy (CD45RA⁺FoxP3⁺)¹⁴⁻²³.

5. Fig. 4. For rigor and reproducibility of the data, the size of the thymic lobules and other important features should be plotted in graphs.

Thymus weights (as a marker of size) were significantly lighter in the preeclamptic group and we provide this information in the supplementary table S5. Detailed macroscopic features such as the size of the thymic lobes were not measured at the time the specimens were collected, as this is not currently recommended as part of a routine autopsy assessment. This would have to be done in a prospective study. Thymic specimens in this study were mainly collected for the purpose of microscopic assessment and performing immunohistochemistry.

6. Fig.5. Not clear why the authors are showing the SCFA concentration data after log-transformed. As it presented, the differences are not clear.

Log transformation of the SCFA concentration data is appropriate as, in a natural scale, the outlying high values contribute excessively to the estimated differences between groups. Log transformation assists in normalising the distribution of these data. Further, log

transformation assists in the interpretation of the findings, as it enables an estimate of the risk ratio per cent change in the SCFA concentration.

7. *There is no evidence presented that acetate normalizes Treg numbers through AIRE regulation. What is the mechanism for AIRE gene regulation by acetate?*

Increased Treg might indeed not be related to increased AIRE expression and we have changed the text accordingly. We also agree with the reviewer that treatment with acetate correlates with increased expression of AIRE, which does not necessarily mean that acetate directly promotes AIRE expression. As noted above, the regulation of AIRE expression is poorly characterised in the current literature. Age, hormones are important factors with aging downregulating AIRE and androgens upregulating its expression. While fascinating, we believe that understanding the mechanisms behind the effects of acetate supplementation on AIRE expression is beyond the scope of this manuscript. The sentence “*However, the exact mechanisms by which acetate enhances Aire and Foxp3 expression and how they may be related, require further investigations.*” has been now added in Discussion on page 17, lines 331-333.

Changes made in the Abstract on page 3, line 51-59: “*In germ-free mice, devoid of gut microbiota, fetal thymus, CD4⁺ T cells and Treg development were similarly compromised but were rescued in the offspring by maternal supplementation with the gut bacterial metabolite short chain fatty acid (SCFA) acetate. Maternal acetate supplementation also led in the offspring thymus to upregulation of the autoimmune regulator (AIRE), which is known to contribute to Treg cell generation. Similarly, in our human cohort, low serum acetate in mothers was associated with decreased maternal and fetal Treg, subsequent preeclampsia, and correlated with low serum acetate in the fetus. These findings suggest a potential role of acetate in the pathogenesis of preeclampsia, and for immune events in offspring.*”

Results Section, pages 13-14, lines 260-276:

“Maternal SCFA acetate supplementation recovers offspring thymic expression levels of the AIRE protein

AIRE contributes to Treg cell generation early in life ²⁴ and Foxp3 levels are important for Treg function ²⁵, therefore we investigated for alterations in Aire expression in thymic

epithelial cells. AIRE is mostly expressed in thymic epithelial cells²⁶. Our gating strategy for flow cytometry was based on the expression of europaeus agglutinin I (UEAI) lectin which discriminates cortical thymic epithelial cells (cTEC; UEAI⁻) from medullary epithelial cells (mTEC; UEAI⁺) while both subsets express major histocompatibility complex II (MHC II) (Fig 7a). The majority of Aire⁺ cells are found in the keratin 14⁺ medulla region (Supplementary Fig S2). However, we also found a minority of Aire⁺ cells with a cortical phenotype ($\beta 5t^+$, keratin 14⁻) situated at the cortico-medullary junction (white arrows, Supplementary Fig S2). We found a significant reduction in the number and proportion of AIRE expressing cells in the cTEC subset in GF mice while AIRE positive cells among the mTEC were unchanged (Fig 7d, e), suggesting a differential role for gut bacterial metabolites on Aire expression at these subsets. The proportion of Aire expressing cTEC was normalised in the offspring of mice supplemented with acetate during pregnancy (Fig 7b-c). Therefore, the impact of acetate on AIRE⁺ cTECs population might contribute to the effect of acetate on Foxp3 expression in Tregs.”

8. Figure 6 shows that CD4 SP T cell numbers are also decreased in GF mice. This indicates that it is not just Tregs that are affected. Despite this, the authors emphasize the changes in Tregs only.

We agree both total CD4⁺ T cells and Treg cells are decreased in GF mice and have added this point in the abstract and discussion. The sentence “In germ free mice, devoid of gut microbiota, fetal thymus, CD4⁺ T cells and Treg development were similarly compromised but were rescued in the offspring by maternal supplementation with the gut bacterial metabolite short chain fatty acid (SCFA) acetate.” has been now added in Abstract on page 3, lines 51-54. The sentence “In a mouse model, we found a key role for maternal microbiota on lymphoid organs and immune cell development. We show that the maternal acetate can affect the development of both thymic and bone marrow in the offspring as well as CD4⁺ T cell development, the expression of AIRE and FoxP3, key components for Treg function.” has been now added in Discussion on page 16, lines 311-314. The reason why we focused more on Treg initially was to be consistent with our findings in humans. As stated previously, unfortunately, we do not have total T cell numbers for the human samples.

9. In methods, they fed mice with an extremely high concentration of sodium acetate (200 mM). This is physiologically irrelevant. In mice, the acetate level in the gut lumen is much

lower than this. How much is required to change SCFA levels in fetal blood and thymus architecture?

Most studies about the impact of acetate on immune function use doses of acetate in drinking water ranging from 150mM to 200mM (^{27,28,29,30,31}). While such doses are not physiological, Smith et al.³² have treated mice with 150mM acetate in water and found that it increased colonic acetate in germ-free mice from 2.82uM to 16.18uM while SPF mice had 40.66uM. This result shows that a high dose of acetate supplementation restores colonic acetate only partially. To improve our manuscript, we did add in the method these references to justify the dose of acetate we used in our study. Studying acetate at different doses and impact on fetal blood acetate and thymic structure is indeed very relevant but beyond the scope of our study.

Reference

- 1 Liston, A. & Gray, D. H. Homeostatic control of regulatory T cell diversity. *Nature reviews. Immunology* 14, 154-165, doi:10.1038/nri3605 (2014).
- 2 Lee, G. R. The Balance of Th17 versus Treg Cells in Autoimmunity. *International journal of molecular sciences* 19, doi:10.3390/ijms19030730 (2018).
- 3 Noack, M. & Miossec, P. Th17 and regulatory T cell balance in autoimmune and inflammatory diseases. *Autoimmun Rev* 13, 668-677, doi:10.1016/j.autrev.2013.12.004 (2014).
- 4 Zhu, X. et al. Correlation of increased Th17/Treg cell ratio with endoplasmic reticulum stress in chronic kidney disease. *Medicine* 97, e10748, doi:10.1097/md.0000000000010748 (2018).
- 5 Hsu, P. & Nanan, R. K. Innate and Adaptive Immune Interactions at the Fetal-Maternal Interface in Healthy Human Pregnancy and Pre-Eclampsia. *Frontiers in Immunology* 5, 125, doi:10.3389/fimmu.2014.00125 (2014).
- 6 Neller, M. A. et al. Multivariate Analysis Using High Definition Flow Cytometry Reveals Distinct T Cell Repertoires between the Fetal-Maternal Interface and the Peripheral Blood. *Frontiers in Immunology* 5, 33, doi:10.3389/fimmu.2014.00033 (2014).
- 7 Santner-Nanan, B. et al. Systemic increase in the ratio between Foxp3+ and IL-17-producing CD4+ T cells in healthy pregnancy but not in preeclampsia. *Journal of Immunology* 183, 7023-7030, doi:10.4049/jimmunol.0901154 (2009).
- 8 Sakaguchi, S., Miyara, M., Costantino, C. M. & Hafler, D. A. FOXP3+ regulatory T cells in the human immune system. *Nat Rev Immunol* 10, 490-500, doi:nri2785 [pii] 10.1038/nri2785 (2010).
- 9 Miyara, M. et al. Functional delineation and differentiation dynamics of human CD4+ T cells expressing the FoxP3 transcription factor. *Immunity* 30, 899-911, doi:10.1016/j.immuni.2009.03.019 (2009).

- 10 Miyara, M. & Sakaguchi, S. Human FoxP3(+)CD4(+) regulatory T cells: their knowns and unknowns. *Immunol Cell Biol* 89, 346-351, doi:icb2010137 [pii] 10.1038/icb.2010.137 (2011).
- 11 Zhang, Y. et al. Cord blood monocyte-derived inflammatory cytokines suppress IL-2 and induce nonclassic "TH2-type" immunity associated with development of food allergy. *Sci Transl Med* 8, 321ra328, doi:10.1126/scitranslmed.aad4322 (2016).
- 12 Collier, F. M. et al. The ontogeny of naive and regulatory CD4(+) T-cell subsets during the first postnatal year: a cohort study. *Clinical & translational immunology* 4, e34, doi:10.1038/cti.2015.2 (2015).
- 13 Collier, F. Naïve regulatory T cells in infancy: Associations with perinatal factors and development of food allergy. *Allergy*.
- 14 Silva-Neta, H. L. et al. CD4(+)CD45RA(-)FOXP3(low) Regulatory T Cells as Potential Biomarkers of Disease Activity in Systemic Lupus Erythematosus Brazilian Patients. *BioMed research international* 2018, 3419565, doi:10.1155/2018/3419565 (2018).
- 15 Mohr, A., Malhotra, R., Mayer, G., Gorochoy, G. & Miyara, M. Human FOXP3(+) T regulatory cell heterogeneity. *Clinical & translational immunology* 7, e1005, doi:10.1002/cti2.1005 (2018).
- 16 Li, Z. et al. Dynamic changes in CD45RA(-)Foxp3(high) regulatory T-cells in chronic hepatitis C patients during antiviral therapy. *International journal of infectious diseases : IJID : official publication of the International Society for Infectious Diseases* 45, 5-12, doi:10.1016/j.ijid.2016.02.006 (2016).
- 17 Zheng, S. G. Regulatory T cells vs Th17: differentiation of Th17 versus Treg, are the mutually exclusive? *American journal of clinical and experimental immunology* 2, 94-106 (2013).
- 18 Saito, S., Shima, T., Inada, K. & Nakashima, A. Which types of regulatory T cells play important roles in implantation and pregnancy maintenance? *American journal of reproductive immunology* 69, 340-345, doi:10.1111/aji.12101 (2013).
- 19 Simonetta, F. et al. Early and long-lasting alteration of effector CD45RA(-)Foxp3(high) regulatory T-cell homeostasis during HIV infection. *The Journal of infectious diseases* 205, 1510-1519, doi:10.1093/infdis/jis235 (2012).
- 20 Schaiyer, M. et al. DR(high+)CD45RA(-)-Tregs potentially affect the suppressive activity of the total Treg pool in renal transplant patients. *PloS one* 7, e34208, doi:10.1371/journal.pone.0034208 (2012).
- 21 Taflin, C. et al. Human endothelial cells generate Th17 and regulatory T cells under inflammatory conditions. *Proceedings of the National Academy of Sciences of the United States of America* 108, 2891-2896, doi:10.1073/pnas.1011811108 (2011).
- 22 Booth, N. J. et al. Different proliferative potential and migratory characteristics of human CD4+ regulatory T cells that express either CD45RA or CD45RO. *Journal of Immunology* 184, 4317-4326, doi:10.4049/jimmunol.0903781 (2010).
- 23 Sahin, E. & Sahin, M. Epigenetical Targeting of the FOXP3 Gene by S-Adenosylmethionine Diminishes the Suppressive Capacity of Regulatory T Cells Ex Vivo and Alters the Expression Profiles. *Journal of immunotherapy* 42, 11-22, doi:10.1097/CJI.0000000000000247 (2019).
- 24 Yang, S., Fujikado, N., Kolodin, D., Benoist, C. & Mathis, D. Immune tolerance. Regulatory T cells generated early in life play a distinct role in maintaining self-tolerance. *Science* 348, 589-594, doi:10.1126/science.aaa7017 (2015).
- 25 Chauhan, S. K., Saban, D. R., Lee, H. K. & Dana, R. Levels of Foxp3 in regulatory T cells reflect their functional status in transplantation. *Journal of Immunology* 182, 148-153 (2009).

- 26 Anderson, M. S. et al. Projection of an Immunological Self Shadow Within the Thymus by the Aire Protein. *Science* 298, 1395-1401, doi:10.1126/science.1075958 (2002).
- 27 Tan, J. et al. Dietary Fiber and Bacterial SCFA Enhance Oral Tolerance and Protect against Food Allergy through Diverse Cellular Pathways. *Cell Reports* 15, 2809-2824, doi:https://doi.org/10.1016/j.celrep.2016.05.047 (2016).
- 28 Macia, L. et al. Metabolite-sensing receptors GPR43 and GPR109A facilitate dietary fibre-induced gut homeostasis through regulation of the inflammasome. *Nature communications* 6, 6734, doi:10.1038/ncomms7734 (2015).
- 29 Thorburn, A. N. et al. Evidence that asthma is a developmental origin disease influenced by maternal diet and bacterial metabolites. *Nature communications* 6, 7320, doi:10.1038/ncomms8320 (2015).
- 30 Vieira, A. T. et al. A Role for Gut Microbiota and the Metabolite-Sensing Receptor GPR43 in a Murine Model of Gout. *Arthritis & rheumatology (Hoboken, N.J.)* 67, 1646-1656, doi:10.1002/art.39107 (2015).
- 31 Maslowski, K. M. et al. Regulation of inflammatory responses by gut microbiota and chemoattractant receptor GPR43. *Nature* 461, 1282-1286, doi:10.1038/nature08530 (2009).
- 32 Smith, P. M. et al. The microbial metabolites, short-chain fatty acids, regulate colonic Treg cell homeostasis. *Science* 341, 569-573, doi:10.1126/science.1241165 (2013).

REVIEWERS' COMMENTS:

Reviewer #1 (Remarks to the Author):

All my concerns have been addressed sufficiently.

Best wishes,
Jakob Stokholm

Reviewer #2 (Remarks to the Author):

The authors revised the manuscript by providing some more information. However, the manuscript still has many weaknesses. More specifically, it is not convincingly shown if acetate produced by microbiota really supports thymus development. It is also not clear it is just Tregs that are supported by the SCFAs. Moreover, there is no data showing the mechanism of AIRE regulation by SCFAs. In methods, the authors used an extremely high concentration of SCFAs to increase SCFA levels in mice, which is supraphysiological and can cause artificial changes in pregnant mice.

A point by point response to the issues raised by the referee has been provided below. We have also toned down our claims within the manuscript as advised by the reviewer and the editor.

1. *Introduction, “we describe a mechanistic link between maternal gut microbial metabolites and fetal immune development in preeclampsia.” Unlike the claim, there is no clear mechanistic data explaining how acetate increased Tregs in eclampsia. More specially, no data are shown to provide mechanisms for increased Tregs and AIRE expression by acetate supplementation. What authors show are simple association of smaller thymus size with preeclampsia. Fetal thymus size is smaller in preeclampsia. This should affect both Tregs and non-Tregs but the authors describe only about Tregs. This is oversimplification of the observed phenomenon.*

We have now changed this section in the introduction as recommended by the referee (page 6, lines 106-109): *“In this study, we describe an association between maternal gut microbial metabolites and fetal immune development in preeclampsia. Specifically, we demonstrate a link between acetate and fetal thymic development and output, with striking differences between non-preeclamptic and preeclamptic pregnancies.”*

“A Link” seems to be an overstatement with the presented data. Decreased thymus size in preeclampsia along with decreased T cell output is presented but this has been largely already described. Acetate increased thymus size in germ-free mice but germ-free mice are not even similar to preeclampsia in humans.

Abstract (page 3, line 58-59): *These findings suggest a potential role of acetate in the pathogenesis of preeclampsia, and for immune events in offspring.*

This makes the manuscript highly speculative without any convincing data that SCFA deficiency causes thymic atrophy in preeclampsia.

Discussion (page 15, line 278-279): *These studies provide a link between maternal gut bacterial metabolite acetate, thymus and Treg development in the context of preeclampsia.*

At best, the authors demonstrated that acetate feeding in germ-free mice increased thymus size, but what is happening in germ-free mice are not likely to indicate what is happening in human preeclampsia.

We agree with the reviewer that treatment with acetate correlates with increased expression of AIRE, which does not necessarily mean that acetate directly promotes AIRE expression. The regulation of AIRE expression is poorly characterised in the current literature. Delineation of the mechanisms behind the effects of acetate supplementation on AIRE expression is beyond the scope of this manuscript. The sentence “*However, the exact mechanisms by which acetate enhances Aire and Foxp3 expression and how they may be related, require further investigations.*” has been now added in Discussion on page 17, lines 331-333.

Again, this weakens the manuscript and make it speculative.

We agree that reduced fetal thymus size could possibly affect both Tregs and non-Tregs; however, in our study, we only found changes in Tregs. This point is addressed in more detail in response to question 3.

This statement does not make sense. Thymus produce all T cells but only Treg numbers were decreased? Perhaps the study has not been performed properly to capture changes in all major T cell subsets as the result of decreased thymus volume.

2. Fig. 1. Impaired fetal thymic development in preeclampsia. Decreased thymus development may be due to smaller body size in preeclampsia. The growth of fetus should be also shown to show the defective thymic development is unique in the fetus.

We agree that fetal thymus size is related to fetal size. To address this, fetal thymus measurements were adjusted for fetal size parameters including estimated fetal weight (Nepean Cohort 1) and fetal head circumference (Nepean Cohort 2). In terms of Fig. 1, we believe that the addition of fetal growth data would unnecessarily clutter the graphs.

This is necessary to support the claims made in the manuscript.

3. Fig. 2. “%Foxp3+ (Mean) within CD4+ cells in blood” are shown in many figures. Thymus produce more non-Tregs than Tregs. Total T cell output should be shown rather than showing only Treg frequency among CD4+ T cells. The frequency may not have to change if total T cell output is down in preeclampsia. The authors should show data for conventional CD4 and CD8 T cell subsets.

We found no evidence that preeclamptic pregnancies were associated with decreased percentages of thymic T cells (CD3 T cells), or conventional CD4/CD8 T cell subsets (see below figure) in contrast to the changes found within the Treg cells compartment. The sentence “*In contrast, we found no evidence that preeclamptic pregnancies were associated with changes in percentages of fetal thymic T cells (CD3 T cells), or conventional CD4/CD8 T cell subsets (data not shown).*” has now been added into Discussion on page 15, lines 294-296.

T cell frequencies would not change but total numbers or thymic out would be decreased in preeclampsia. Without this information, the data are misleading.

There is compelling data suggesting that changes in percentages of Treg cells within CD4⁺ cells play a central role in many immune-mediated pathologies¹⁻⁴. In line with this, our own group has shown reduced percentages in maternal Tregs to be a feature of altered immunity

in preeclampsia⁵⁻⁷. Hence, differences in percentages of offspring Treg cells described in this study between the preeclampsia and non-preeclampsia groups seems highly relevant.

We have no data on the absolute (total) numbers of thymic Treg cells or CD4/CD8 T cell subsets. To calculate total numbers of these subsets would have required the availability of full blood counts, which were not systematically collected for any of the cohorts.

4. Fig. 3. The simple gating based on CD45RA and FoxP3 is not a reliable way to identify Treg subsets due to rather faint staining of FoxP3. This is prone to include non-Tregs. Utilization of other antigens such as CD25, CTLA4 and others should help with the identification.

Our approach was based on the seminal studies of Shimon Sakaguchi⁸⁻¹⁰ who showed that “the combination of FoxP3 and CD45RA expression can dissect FoxP3⁺ cells into three subpopulations: (i) CD45RA⁺FoxP3^{low} naive Tregs (nTregs); (ii) CD45RA⁺FoxP3^{high} effector Tregs (eTregs), both of which are potently suppressive in vitro; and (iii) non-suppressive, cytokine secreting CD45RA⁺FoxP3^{low} non-Tregs”¹⁰.

Sakaguchi et al also state that “As the expression levels of FoxP3 are proportional to those of CD25, the use of CD25 and CD45RA enables the isolation of the three populations as live cells”¹⁰. In other words, CD25 can be used as a surface marker for isolation of individual populations of Tregs, but is not necessary in combination with FoxP3 for the assessment and discrimination of nTreg.

As shown in Figure 3 the degree of FoxP3 staining was always strong enough to discriminate FoxP3-positive from FoxP3-negative cells. Finally, we have published previously on the nTreg measured in this way¹¹⁻¹³, and there are many other publications that have used the same strategy (CD45RA⁺FoxP3⁺)¹⁴⁻²³.

5. Fig. 4. For rigor and reproducibility of the data, the size of the thymic lobules and other important features should be plotted in graphs.

Thymus weights (as a marker of size) were significantly lighter in the preeclamptic group and we provide this information in the supplementary table S5. Detailed macroscopic features such as the size of the thymic lobes were not measured at the time the specimens were collected, as this is not currently recommended as part of a routine autopsy assessment. This would have to be done in a prospective study. Thymic specimens in this study were mainly collected for the purpose of microscopic assessment and performing immunohistochemistry.

6. Fig.5. Not clear why the authors are showing the SCFA concentration data after log-transformed. As it presented, the differences are not clear.

Log transformation of the SCFA concentration data is appropriate as, in a natural scale, the outlying high values contribute excessively to the estimated differences between groups. Log transformation assists in normalising the distribution of these data. Further, log transformation assists in the interpretation of the findings, as it enables an estimate of the risk ratio per cent change in the SCFA concentration.

The log transformation is confusing. The authors should show actual concentrations of SCFAs to be informative.

7. There is no evidence presented that acetate normalizes Treg numbers through AIRE regulation. What is the mechanism for AIRE gene regulation by acetate?

Increased Treg might indeed not be related to increased AIRE expression and we have changed the text accordingly. We also agree with the reviewer that treatment with acetate correlates with increased expression of AIRE, which does not necessarily mean that acetate directly promotes AIRE expression. As noted above, the regulation of AIRE expression is poorly characterised in the current literature. Age, hormones are important factors with aging downregulating AIRE and androgens upregulating its expression. While fascinating, we believe that understanding the mechanisms behind the effects of acetate supplementation on AIRE expression is beyond the scope of this manuscript. The sentence “*However, the exact mechanisms by which acetate enhances Aire and Foxp3 expression and how they may be*

related, require further investigations.” has been now added in Discussion on page 17, lines 331-333.

Again, without the data, the message is only speculative without any hard evidence.

Changes made in the Abstract on page 3, line 51-59: *“In germ-free mice, devoid of gut microbiota, fetal thymus, CD4⁺ T cells and Treg development were similarly compromised but were rescued in the offspring by maternal supplementation with the gut bacterial metabolite short chain fatty acid (SCFA) acetate. Maternal acetate supplementation also led in the offspring thymus to upregulation of the autoimmune regulator (AIRE), which is known to contribute to Treg cell generation. Similarly, in our human cohort, low serum acetate in mothers was associated with decreased maternal and fetal Treg, subsequent preeclampsia, and correlated with low serum acetate in the fetus. These findings suggest a potential role of acetate in the pathogenesis of preeclampsia, and for immune events in offspring.”*

Results Section, pages 13-14, lines 260-276:

“Maternal SCFA acetate supplementation recovers offspring thymic expression levels of the AIRE protein

AIRE contributes to Treg cell generation early in life²⁴ and Foxp3 levels are important for Treg function²⁵, therefore we investigated for alterations in Aire expression in thymic epithelial cells. AIRE is mostly expressed in thymic epithelial cells²⁶. Our gating strategy for flow cytometry was based on the expression of europaeus agglutinin 1 (UEA1) lectin which discriminates cortical thymic epithelial cells (cTEC; UEA1⁻) from medullary epithelial cells (mTEC; UEA1⁺) while both subsets express major histocompatibility complex II (MHC II) (Fig 7a). The majority of Aire⁺ cells are found in the keratin 14⁺ medulla region (Supplementary Fig S2). However, we also found a minority of Aire⁺ cells with a cortical phenotype ($\beta 5t^+$, keratin 14) situated at the cortico-medullary junction (white arrows, Supplementary Fig S2). We found a significant reduction in the number and proportion of AIRE expressing cells in the cTEC subset in GF mice while AIRE positive cells among the mTEC were unchanged (Fig 7d, e), suggesting a differential role for gut bacterial metabolites on Aire expression at these subsets. The proportion of Aire expressing cTEC was

normalised in the offspring of mice supplemented with acetate during pregnancy (Fig 7b-c). Therefore, the impact of acetate on AIRE⁺ cTECs population might contribute to the effect of acetate on Foxp3 expression in Tregs.”

8. Figure 6 shows that CD4 SP T cell numbers are also decreased in GF mice. This indicates that it is not just Tregs that are affected. Despite this, the authors emphasize the changes in Tregs only.

We agree both total CD4⁺ T cells and Treg cells are decreased in GF mice and have added this point in the abstract and discussion. The sentence “*In germ free mice, devoid of gut microbiota, fetal thymus, CD4⁺ T cells and Treg development were similarly compromised but were rescued in the offspring by maternal supplementation with the gut bacterial metabolite short chain fatty acid (SCFA) acetate.*” has been now added in Abstract on page 3, lines 51-54. The sentence “*In a mouse model, we found a key role for maternal microbiota on lymphoid organs and immune cell development. We show that the maternal acetate can affect the development of both thymic and bone marrow in the offspring as well as CD4⁺ T cell development, the expression of AIRE and FoxP3, key components for Treg function.*” has been now added in Discussion on page 16, lines 311-314. The reason why we focused more on Treg initially was to be consistent with our findings in humans. As stated previously, unfortunately, we do not have total T cell numbers for the human samples.

9. In methods, they fed mice with an extremely high concentration of sodium acetate (200 mM). This is physiologically irrelevant. In mice, the acetate level in the gut lumen is much lower than this. How much is required to change SCFA levels in fetal blood and thymus architecture?

Most studies about the impact of acetate on immune function use doses of acetate in drinking water ranging from 150mM to 200mM (^{27,28,29,30,31}). While such doses are not physiological, Smith et al.³² have treated mice with 150mM acetate in water and found that it increased colonic acetate in germ-free mice from 2.82uM to 16.18uM while SPF mice had 40.66uM. This result shows that a high dose of acetate supplementation restores colonic acetate only partially. To improve our manuscript, we did add in the method these references to justify the dose of acetate we used in our study. Studying acetate at different doses and impact on fetal blood acetate and thymic structure is indeed very relevant but beyond the scope of our study.

In early studies, high concentrations of C2 were used, but it has been later reported that inflammatory responses can occur when feeding mice with SCFAs at higher than 150 mM. This can make the interpretations difficult.

Dear Editor,

Thank you for giving us the opportunity to respond to both reviewer's remaining concerns.

Reviewer #1 (Remarks to the Author):

All my concerns have been addressed sufficiently.

Best wishes,

Jakob Stokholm

Reviewer #2 (Remarks to the Author):

Note: initial response in black, reviewer comment in red, and our subsequent response in blue.

1. Introduction, “we describe a mechanistic link between maternal gut microbial metabolites and fetal immune development in preeclampsia.” Unlike the claim, there is no clear mechanistic data explaining how acetate increased Tregs in eclampsia. More specially, no data are shown to provide mechanisms for increased Tregs and AIRE expression by acetate supplementation. What authors show are simple association of smaller thymus size with preeclampsia. Fetal thymus size is smaller in preeclampsia. This should affect both Tregs and non-Tregs but the authors describe only about Tregs. This is oversimplification of the observed phenomenon.

We have now changed this section in the introduction as recommended by the referee (page 6, lines 106-109): *“In this study, we describe an association between maternal gut microbial metabolites and fetal immune development in preeclampsia. Specifically, we demonstrate a link between acetate and fetal thymic development and output, with striking differences between non-preeclamptic and preeclamptic pregnancies.”*

“A Link” seems to be an overstatement with the presented data. Decreased thymus size in preeclampsia along with decreased T cell output is presented but this has been largely already described. Acetate increased thymus size in germ-free mice but germ-free mice are not even similar to preeclampsia in humans.

Response: The constellation of evidence in our paper provides a cogent basis for future experimental studies in humans.

We have amended this statement to delineate between the observational findings in humans and the experimental findings in mice (page 6, lines 101-105): *“In this study, we present evidence of an association between maternal gut microbial metabolites and fetal immune development in preeclampsia. Specifically, we demonstrate in mice a link between acetate and fetal thymic development and output, with concordant associations in human cohorts*

between decreased serum acetate and subsequent preeclampsia, and between preeclampsia and decreased thymic size and output in the offspring.”

Abstract (page 3, line 58-59): *These findings suggest a potential role of acetate in the pathogenesis of preeclampsia, and for immune events in offspring.*

This makes the manuscript highly speculative without any convincing data that SCFA deficiency causes thymic atrophy in preeclampsia.

Our findings do suggest a potential role of acetate in the pathogenesis of preeclampsia and fetal immune development. We believe the current wording is appropriate.

Discussion (page 15, line 278-279): *These studies provide a link between maternal gut bacterial metabolite acetate, thymus and Treg development in the context of preeclampsia.*

At best, the authors demonstrated that acetate feeding in germ-free mice increased thymus size, but what is happening in germ-free mice are not likely to indicate what is happening in human preeclampsia.

We have added the word ‘plausible’ to this sentence. Further experimental studies in humans are required to confirm our findings. Discussion (page 15, lines 276-277): “These studies provide a plausible link between maternal gut bacterial metabolite acetate, thymus and Treg development in the context of preeclampsia.”

We agree with the reviewer that treatment with acetate correlates with increased expression of AIRE, which does not necessarily mean that acetate directly promotes AIRE expression. The regulation of AIRE expression is poorly characterised in the current literature. Delineation of the mechanisms behind the effects of acetate supplementation on AIRE expression is beyond the scope of this manuscript. The sentence “*However, the exact mechanisms by which acetate enhances Aire and Foxp3 expression and how they may be related, require further investigations.*” has been now added in Discussion on page 17, lines 331-333.

Again, this weakens the manuscript and make it speculative.

We believe our interpretation of the evidence presented regarding AIRE is not overstated, and appropriate caveats provided.

We agree that reduced fetal thymus size could possibly affect both Tregs and non-Tregs; however, in our study, we only found changes in Tregs. This point is addressed in more detail in response to question 3.

This statement does not make sense. Thymus produce all T cells but only Treg numbers were decreased? Perhaps the study has not been performed properly to capture changes in all major T cell subsets as the result of decreased thymus volume.

Our statement simply reflects what was apparent in our data. Other cells lines were investigated but we only observed changes in Treg. This was true of both the observational findings from human cohorts and the experimental findings in mice.

2. Fig. 1. Impaired fetal thymic development in preeclampsia. Decreased thymus development may be due to smaller body size in preeclampsia. The growth of fetus should be also shown to show the defective thymic development is unique in the fetus.

We agree that fetal thymus size is related to fetal size. To address this, fetal thymus measurements were adjusted for fetal size parameters including estimated fetal weight (Nepean Cohort 1) and fetal head circumference (Nepean Cohort 2). In terms of Fig. 1, we believe that the addition of fetal growth data would unnecessarily clutter the graphs.

This is necessary to support the claims made in the manuscript.

3. Fig. 2. “%Foxp3+ (Mean) within CD4+ cells in blood” are shown in many figures. Thymus produce more non-Tregs than Tregs. Total T cell output should be shown rather than showing only Treg frequency among CD4+ T cells. The frequency may not have to change if total T cell output is down in preeclampsia. The authors should show data for conventional CD4 and CD8 T cell subsets.

We found no evidence that preeclamptic pregnancies were associated with decreased percentages of thymic T cells (CD3 T cells), or conventional CD4/CD8 T cell subsets (see below figure) in contrast to the changes found within the Treg cells compartment. The sentence “*In contrast, we found no evidence that preeclamptic pregnancies were associated with changes in percentages of fetal thymic T cells (CD3 T cells), or conventional CD4/CD8 T cell subsets (data not shown).*” has now been added into Discussion on page 15, lines 294-296.

There is compelling data suggesting that changes in percentages of Treg cells within CD4⁺ cells play a central role in many immune-mediated pathologies¹⁻⁴. In line with this, our own group has shown reduced percentages in maternal Tregs to be a feature of altered immunity in preeclampsia⁵⁻⁷. Hence, differences in percentages of offspring Treg cells described in this study between the preeclampsia and non-preeclampsia groups seems highly relevant.

We have no data on the absolute (total) numbers of thymic Treg cells or CD4/CD8 T cell subsets. To calculate total numbers of these subsets would have required the availability of full blood counts, which were not systematically collected for any of the cohorts.

T cell frequencies would not change but total numbers or thymic out would be decreased in preeclampsia. Without this information, the data are misleading.

Additional sentences “We have no data on the absolute (total) numbers of thymic Treg cells or CD4/CD8 T cell subsets. To calculate total numbers of these subsets would have required

the availability of full blood counts, which were not systematically collected for any of the cohorts. However, there is compelling data suggesting that changes in percentages of Treg cells within CD4⁺ cells play a central role in many immune-mediated pathologies¹⁻⁴. In line with this, our own group has shown reduced percentages in maternal Tregs to be a feature of altered immunity in preeclampsia⁵⁻⁷. Hence, differences in percentages of offspring Treg cells described in this study between the preeclampsia and non-preeclampsia groups seems highly relevant.” have been added now to page 15-16, lines 294-301.

The figure showing “preeclamptic pregnancies were associated with decreased percentages of thymic T cells (CD3 T cells), or conventional CD4/CD8 T cell subsets” has also been added as Supplementary figure 6 now.

6. Fig.5. Not clear why the authors are showing the SCFA concentration data after log-transformed. As it presented, the differences are not clear.

Log transformation of the SCFA concentration data is appropriate as, in a natural scale, the outlying high values contribute excessively to the estimated differences between groups. Log transformation assists in normalising the distribution of these data. Further, log transformation assists in the interpretation of the findings, as it enables an estimate of the risk ratio per cent change in the SCFA concentration.

The log transformation is confusing. The authors should show actual concentrations of SCFAs to be informative.

We have sought extensive biostatistical input on this question and we are confident that, for the reasons noted, presentation in a log scale is appropriate.

7. There is no evidence presented that acetate normalizes Treg numbers through AIRE regulation. What is the mechanism for AIRE gene regulation by acetate?

Increased Treg might indeed not be related to increased AIRE expression and we have changed the text accordingly. We also agree with the reviewer that treatment with acetate correlates with increased expression of AIRE, which does not necessarily mean that acetate directly promotes AIRE expression. As noted above, the regulation of AIRE expression is poorly characterised in the current literature. Age, hormones are important factors with aging downregulating AIRE and androgens upregulating its expression. While fascinating, we believe that understanding the mechanisms behind the effects of acetate supplementation on AIRE expression is beyond the scope of this manuscript. The sentence “*However, the exact mechanisms by which acetate enhances Aire and Foxp3 expression and how they may be related, require further investigations.*” has been now added in Discussion on page 17, lines 331-333.

Again, without the data, the message is only speculative without any hard evidence.

We totally agree that our results are correlative and not hard evidence. However, we still think that showing that acetate promotes AIRE expression will be relevant for the scientific community in general as well as for experts on Treg. It also opens up a new dimension on how SCFA might regulate the T cell repertoire by deleting autoreactive T cells through regulation of AIRE which should also be fully demonstrated. The need for further investigation is acknowledged.

Changes made in the Abstract on page 3, line 51-59: “*In germ-free mice, devoid of gut microbiota, fetal thymus, CD4⁺ T cells and Treg development were similarly compromised but were rescued in the offspring by maternal supplementation with the gut bacterial metabolite short chain fatty acid (SCFA) acetate. Maternal acetate supplementation also led in the offspring thymus to upregulation of the autoimmune regulator (AIRE), which is known to contribute to Treg cell generation. Similarly, in our human cohort, low serum acetate in mothers was associated with decreased maternal and fetal Treg, subsequent preeclampsia, and correlated with low serum acetate in the fetus. These findings suggest a potential role of acetate in the pathogenesis of preeclampsia, and for immune events in offspring.*”

Results Section, pages 13-14, lines 260-276:

“*Maternal SCFA acetate supplementation recovers offspring thymic expression levels of the AIRE protein*”

AIRE contributes to Treg cell generation early in life⁸ and Foxp3 levels are important for Treg function⁹, therefore we investigated for alterations in Aire expression in thymic epithelial cells. AIRE is mostly expressed in thymic epithelial cells¹⁰. Our gating strategy for flow cytometry was based on the expression of europaeus agglutinin I (UEAI) lectin which discriminates cortical thymic epithelial cells (cTEC; UEAI⁻) from medullary epithelial cells (mTEC; UEAI⁺) while both subsets express major histocompatibility complex II (MHC II) (Fig 7a). The majority of Aire⁺ cells are found in the keratin 14⁺ medulla region (Supplementary Fig S2). However, we also found a minority of Aire⁺ cells with a cortical phenotype ($\beta 5t^+$, keratin 14⁻) situated at the cortico-medullary junction (white arrows, Supplementary Fig S2). We found a significant reduction in the number and proportion of AIRE expressing cells in the cTEC subset in GF mice while AIRE positive cells among the mTEC were unchanged (Fig 7d, e), suggesting a differential role for gut bacterial metabolites on Aire expression at these subsets. The proportion of Aire expressing cTEC was normalised in the offspring of mice supplemented with acetate during pregnancy (Fig 7b-c). Therefore, the impact of acetate on AIRE⁺ cTECs population might contribute to the effect of acetate on Foxp3 expression in Tregs.”

9. In methods, they fed mice with an extremely high concentration of sodium acetate (200 mM). This is physiologically irrelevant. In mice, the acetate level in the gut lumen is much lower than this. How much is required to change SCFA levels in fetal blood and thymus architecture?

Most studies about the impact of acetate on immune function use doses of acetate in drinking water ranging from 150mM to 200mM (^{11,12,13,14,15}). While such doses are not physiological, Smith et al.¹⁶ have treated mice with 150mM acetate in water and found that it increased colonic acetate in germ-free mice from 2.82uM to 16.18uM while SPF mice had 40.66uM. This result shows that a high dose of acetate supplementation restores colonic acetate only partially. To improve our manuscript, we did add in the method these references to justify the dose of acetate we used in our study. Studying acetate at different doses and impact on fetal blood acetate and thymic structure is indeed very relevant but beyond the scope of our study.

In early studies, high concentrations of C2 were used, but it has been later reported that inflammatory responses can occur when feeding mice with SCFAs at higher than 150 mM. This can make the interpretations difficult.

We have demonstrated in three studies done in different inflammatory mouse models that when we provided acetate at 200mM in drinking water to for example NOD mice for 25 weeks, mice were protected from diabetes correlating with decrease inflammation in the pancreas¹⁷. Similar approach was beneficial in the colitis model¹² and in asthma¹³.

References

- 1 Liston, A. & Gray, D. H. Homeostatic control of regulatory T cell diversity. *Nature reviews. Immunology* **14**, 154-165, doi:10.1038/nri3605 (2014).
- 2 Lee, G. R. The Balance of Th17 versus Treg Cells in Autoimmunity. *International journal of molecular sciences* **19**, doi:10.3390/ijms19030730 (2018).
- 3 Noack, M. & Miossec, P. Th17 and regulatory T cell balance in autoimmune and inflammatory diseases. *Autoimmun Rev* **13**, 668-677, doi:10.1016/j.autrev.2013.12.004 (2014).
- 4 Zhu, X. *et al.* Correlation of increased Th17/Treg cell ratio with endoplasmic reticulum stress in chronic kidney disease. *Medicine* **97**, e10748, doi:10.1097/md.0000000000010748 (2018).
- 5 Hsu, P. & Nanan, R. K. Innate and Adaptive Immune Interactions at the Fetal-Maternal Interface in Healthy Human Pregnancy and Pre-Eclampsia. *Frontiers in immunology* **5**, 125, doi:10.3389/fimmu.2014.00125 (2014).
- 6 Neller, M. A. *et al.* Multivariate Analysis Using High Definition Flow Cytometry Reveals Distinct T Cell Repertoires between the Fetal-Maternal Interface and the Peripheral Blood. *Frontiers in immunology* **5**, 33, doi:10.3389/fimmu.2014.00033 (2014).
- 7 Santner-Nanan, B. *et al.* Systemic increase in the ratio between Foxp3+ and IL-17-producing CD4+ T cells in healthy pregnancy but not in preeclampsia. *Journal of immunology* **183**, 7023-7030, doi:10.4049/jimmunol.0901154 (2009).
- 8 Yang, S., Fujikado, N., Kolodin, D., Benoist, C. & Mathis, D. Immune tolerance. Regulatory T cells generated early in life play a distinct role in maintaining self-tolerance. *Science* **348**, 589-594, doi:10.1126/science.aaa7017 (2015).
- 9 Chauhan, S. K., Saban, D. R., Lee, H. K. & Dana, R. Levels of Foxp3 in regulatory T cells reflect their functional status in transplantation. *Journal of immunology* **182**, 148-153 (2009).
- 10 Anderson, M. S. *et al.* Projection of an Immunological Self Shadow Within the Thymus by the Aire Protein. *Science* **298**, 1395-1401, doi:10.1126/science.1075958 (2002).
- 11 Tan, J. *et al.* Dietary Fiber and Bacterial SCFA Enhance Oral Tolerance and Protect against Food Allergy through Diverse Cellular Pathways. *Cell Reports* **15**, 2809-2824, doi:<https://doi.org/10.1016/j.celrep.2016.05.047> (2016).

- 12 Macia, L. *et al.* Metabolite-sensing receptors GPR43 and GPR109A facilitate dietary fibre-induced gut homeostasis through regulation of the inflammasome. *Nature communications* **6**, 6734, doi:10.1038/ncomms7734 (2015).
- 13 Thorburn, A. N. *et al.* Evidence that asthma is a developmental origin disease influenced by maternal diet and bacterial metabolites. *Nature communications* **6**, 7320, doi:10.1038/ncomms8320 (2015).
- 14 Vieira, A. T. *et al.* A Role for Gut Microbiota and the Metabolite-Sensing Receptor GPR43 in a Murine Model of Gout. *Arthritis & rheumatology (Hoboken, N.J.)* **67**, 1646-1656, doi:10.1002/art.39107 (2015).
- 15 Maslowski, K. M. *et al.* Regulation of inflammatory responses by gut microbiota and chemoattractant receptor GPR43. *Nature* **461**, 1282-1286, doi:10.1038/nature08530 (2009).
- 16 Smith, P. M. *et al.* The microbial metabolites, short-chain fatty acids, regulate colonic Treg cell homeostasis. *Science* **341**, 569-573, doi:10.1126/science.1241165 (2013).
- 17 Marino, E. *et al.* Gut microbial metabolites limit the frequency of autoimmune T cells and protect against type 1 diabetes. *Nature immunology* **18**, 552-562, doi:10.1038/ni.3713 (2017).